



# GemPy 1.0: open-source stochastic geological modeling and inversion

Miguel de la Varga, Alexander Schaaf, and Florian Wellmann

Institute for Computational Geoscience and Reservoir Engineering

*Correspondence to:* varga@aices.rwth-aachen.de

**Abstract.** The representation of subsurface structures is an essential aspect of a wide variety of geoscientific investigations and applications: ranging from geofluid reservoir studies, over raw material investigations, to geosequestration, as well as many branches of geoscientific research studies and applications in geological surveys. A wide range of methods exists to generate geological models. However, especially the powerful methods are behind a paywall in expensive commercial packages. We present here a full open-source geomodeling method, based on an implicit potential-field interpolation approach. The interpolation algorithm is comparable to implementations in commercial packages and capable of constructing complex full 3-D geological models, including fault networks, fault-surface interactions, unconformities, and dome structures. This algorithm is implemented in the programming language Python, making use of a highly efficient underlying library for efficient code generation (theano) that enables a direct execution on GPU's. The functionality can be separated into the core aspects required to generate 3-D geological models and additional assets for advanced scientific investigations. These assets provide the full power behind our approach, as they enable the link to Machine Learning and Bayesian inference frameworks and thus a path to stochastic geological modeling and inversions. In addition, we provide methods to analyse model topology and to compute gravity fields on the basis of the geological models and assigned density values. In summary, we provide a basis for open scientific research using geological models, with the aim to foster reproducible research in the field of geomodeling.

## Contents





## 1 Introduction

We commonly capture our knowledge about relevant geological features in the subsurface in the form
of geological models, as 3-D representations of the geometric structural setting. Computer-aided geo
logical modeling methods have existed for decades, and many advanced and elaborate commercial
packages exist to generate these models (e.g. GoCAD, Petrel, GeoModeller). But even though these





packages partly enable an external access to the modeling functionality through implemented API's
or scripting interfaces, it is a significant disadvantage that the source code is not accessible, and
therefore the true inner workings are not clear. More importantly still, the possibility to extend these
methods is limited—and, especially in the current rapid development of highly efficient open-source
libraries for machine learning and computational inference (e.g. *TensorFlow*, *Stan*, *pymc*, *PyTorch*,
*Infer.NET*), the integration into other computational frameworks is limited.

Yet, there is to date no fully flexible open-source project which integrates state-of-the-art geo-
logical modeling methods. Conventional 3-D construction tools (CAD, e.g. *pythonOCC*, *PyGem*)
are only useful to a limited extent, as they do not consider the specific aspects of subsurface struc-
tures and the inherent sparcity of data. Open source GIS tools exist (e.g. QGIS, *gdal*), but they are
typically limited to 2-D (or 2.5-D) structures and do not facilitate the modeling and representation
of fault networks, complex structures like overturned folds or dome structures), or combined strati-
graphic sequences.

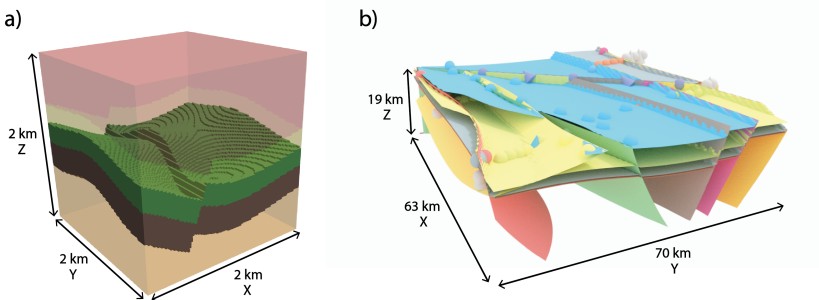

**Figure 1.** Example of models generated using *GemPy*. a) Synthetic model representing a reservoir trap, visual-
ized in Paraview (Stamm, 2017); b) Geological model of the Perth basin (Australia) rendered using GemPy on
the in-built Python in Blender (see appendix F for more details), spheres and cones represent the input data.

With the aim to close this gap, we present here *GemPy*, an open-source implementation of a mod-
ern and powerful implicit geological modeling method based on a potential-field approach, found,
in turn, on a Universal CoKriging interpolation (Lajaunie et al., 1997; Calcagno et al., 2008). In
distinction to surface-based modeling approaches (see Caumon et al., 2009, for a good overview),
these approaches allow the direct interpolation of multiple conformal sequences in a single scalar
field, and the consideration of discontinuities (e.g. metamorphic contacts, unconformities) through
the interaction of multiple sequences (Lajaunie et al., 1997; Mallet, 2004; Calcagno et al., 2008;
Caumon, 2010; Hillier et al., 2014). Also, these methods allow the construction of complex fault
networks and enable, in addition, a direct global interpolation of all available geological data in a
single step. This last aspect is relevant, as it facilitates the integration of these methods into diverse
other workflows. Most importantly, we show here how we can integrate the method into novel and





advanced machine learning and Bayesian inference frameworks (Salvatier et al., 2016) for stochastic geomodeling and Bayesian inversion. Recent developments in this field have seen a surge in new methods and frameworks, for example using gradient-based Monte Carlo methods (Duane et al.,

1987; Hoffman and Gelman, 2014) or variational inferences (Kucukelbir et al., 2016), making use of efficient implementations of automatic differentiation (Rall, 1981) in novel machine learning frameworks. For this reason, *GemPy* is built on top of *Theano*, which provides not only the mentioned capacity to compute gradients efficiently, but also provides optimized compiled code (for more details see Section 2.3.2). In addition, we utilize *pandas* for data storage and manipulation (McKinney,

2011), Visualization Toolkit (*vtk*) Python-bindings for interactive 3-D visualization (Schroeder et al., 2004), the de facto standard 2-D visualization library *Matplotlib* (Hunter, 2007) and *NumPy* for efficient numerical computations (Walt et al., 2011). Our implementation is specifically intended for combination with other packages, to harvest efficient implementations in the best possible way.

Especially in this current time of rapid developments of open-source scientific software packages

and powerful machine learning frameworks, we consider an open-source implementation of a geological modeling tool as essential. We therefore aim to open up this possibility to a wide community, by combining state-of-the-art implicit geological modeling techniques with additional sophisticated Python packages for scientific programming and data analysis in an open-source ecosystem. The aim is explicitly not to rival the existing commercial packages with well-designed graphical user

interfaces, underlying databases, and highly advanced workflows for specific tasks in subsurface engineering, but to provide access to an advanced modeling algorithm for scientific experiments in the field of geomodeling.

In the following, we will present the implementation of our code in the form of core modules, related to the task of geological modeling itself, and additional assets, which provide the link to ex-

ternal libraries, for example to facilitate stochastic geomodeling and the inversion of structural data. Each part is supported/ supplemented with Jupyter Notebooks that are available as additional online material and part of the package documentation, which enable the direct testing of our methods (see Section A3). These notebooks can also be executed directly in an online environment (Binder). We encourage the reader to use these tutorial Jupyter Notebooks to follow along the steps explained in

the following. We encourage the reader to use these tutorial Jupyter Notebooks to follow along the steps explained in the following. Finally, we discuss our approach, specifically also with respect to alternative modeling approaches in the field, and provide an outlook to our planned future work for this project.

## 2 CORE – Geological modeling with GemPy

In this section, we describe the core functionality of *GemPy*: the construction of 3-D geological models from geological input data (surface contact points and orientation measurements) and defined



topological relationships (stratigraphic sequences and fault networks). We begin with a brief review of the theory underlying the implemented interpolation algorithm. We then describe the translation of this algorithm and the subsequent model generation and visualisation using the Python front-end of *GemPy* and how an entire model can be constructed by calling only a few functions. Across the text, we include code snippets with minimal working examples to demonstrate the use of the library.

After describing the simple functionality required to construct models, we go deeper into the underlying architecture of *GemPy*. This part is not only relevant for advanced users and potential developers, but also highlights a key aspect: the link to *Theano* (Theano Development Team, 2016), a highly evolved Python library for efficient vector algebra and machine learning, which is an essential aspect required for making use of the more advanced aspects of stochastic geomodeling and Bayesian inversion, which will also be explained in the subsequent sections.

### 2.1 Geological modeling and the potential-field approach

#### 2.1.1 Concept of the potential-field method

The potential-field method developed by Lajaunie et al. (1997) is the central method to generate the 3D geological models in *GemPy*, which has already been successfully deployed in the modeling software GeoModeller 3-D (see Calcagno et al., 2008). The general idea is to construct an interpolation function $\mathbf{Z}(\mathbf{x}_0)$ where x is any point in the continuous three-dimensional space $(x, y, z) \in \mathbb{R}^3$ which describes the domain $\mathcal{D}$ as a scalar field. The gradient of the scalar field will follow the direction of the anisotropy of the stratigraphic structure or, in other words, every possible isosurface of the scalar field will represent every synchronal deposition of the layer (see figure 2).

Let's break down what we actually mean by this: Imagine that a geological setting is formed by a perfect sequence of horizontal layers piled one above the other. If we know the exact timing of when one of these surfaces was deposited, we would know that any layer above had to occur afterwards while any layer below had to be deposited earlier in time. Obviously, we cannot have data for each of these infinitesimal synchronal layers, but we can interpolate the "date" between them. In reality, the exact year of the synchronal deposition is meaningless—since the related uncertainty would be out of proportion. What has value to generate a 3D geomodel is the location of those synchronal layers and especially the lithological interfaces where the change of physical properties are notable. Due to this, instead interpolating *time*, we use a simple dimensionless parameter—that we simply refer to as *scalar field value*.

The advantages of using a global interpolator instead of interpolating each layer of interest independently are twofold: (i) the location of one layer affects the location of others in the same depositional environment, making impossible for two layers in the same potential field to cross; and (ii) it enables the use of data in-between the interfaces of interest, opening the range of possible measurements that can be used in the interpolation.





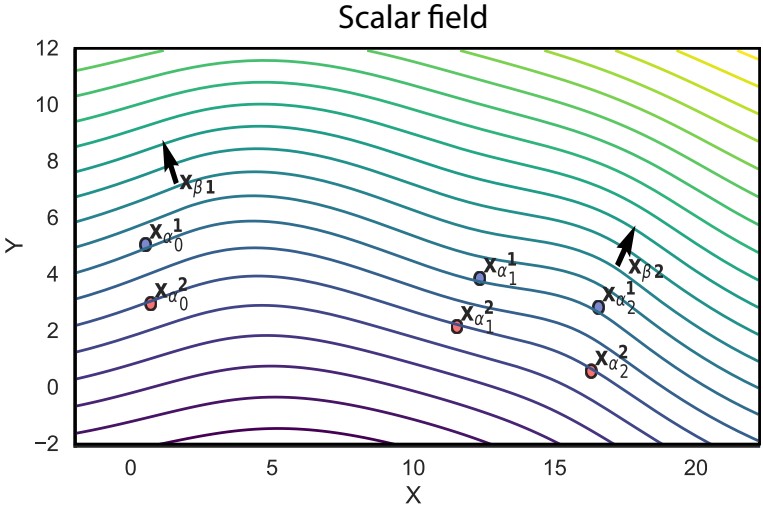

**Figure 2.** Example of scalar field. The input data is formed by six points distributed in two layers ($\mathbf{x}_{\alpha\,i}^1$ and $\mathbf{x}_{\alpha\,i}^2$) and and two orientations ($\mathbf{x}_{\beta\,j}$). A isosurface connect the interface points and the scalar field is perpendicular to the foliation gradient.

The interpolation function is obtained as a weighted interpolation based on Universal CoKriging (Chiles and Delfiner, 2009). Kriging or Gaussian process regression (Matheron, 1981) is a spatial interpolation that treats each input as a random variable, aiming to minimize the covariance function to obtain the best linear unbiased predictor (for a detailed description see Chapter 3 in Wackernagel, 2013). Furthermore, it is possible to combine more than one type of data—i.e. a multivariate case or CoKriging—to increase the amount of information in the interpolator, as long as we capture their relation using a cross-covariance. The main advantage in our case is to be able to utilize data sampled from different locations in space for the estimation. Simple Kriging, as a regression, only minimizes the second moment of the data (or variances). However in most geological settings, we can expect linear trends in our data—i.e. the mean thickness of a layer varies across the region linearly. This trend is captured using polynomial drift functions to the system of equations in what is called Universal Kriging.

### 2.1.2 Adjustments to structural geological modeling

So far we have shown what we want to obtain and how Universal CoKriging is a suitable interpolation method to get there. In the following, we will describe the concrete steps from taking our input data to the final interpolation function $\mathbf{Z}(\mathbf{x}_0)$, which describes the domain. Much of the complexity of the method comes from the difficulty of keeping highly nested nomenclature consistent across literature. For this reason, we will try to be especially verbose regarding the mathematical terminol-



ogy. The terms of *potential field* (original coined by Lajaunie et al., 1997) and *scalar field* (preferred by the authors) are used interchangeably across the paper. The result of a Kriging interpolation is a random function and hence both *interpolation function* and *random function* are used to refer the function of interest $\mathbf{Z}(\mathbf{x}_0)$. The CoKriging nomenclature quickly grows complicated, since it has to consider $p$ random functions $\mathbf{Z}_i$, with $p$ being the number of distinct parameters involved in the inter-

polation, sampled at different points $\mathbf{x}$ of the three-dimensional domain $\mathbb{R}^3$. Two types of parameters are used to characterize the *scalar field* in the interpolation: (i) layer interface points $\mathbf{x}_\alpha$ describing the respective isosurfaces of interest—usually the interface between two layers; and (ii) the gradients of the scalar field, $\mathbf{x}_\beta$—or in geological terms: poles of the layer, i.e. normal vectors to the dip plane. Therefore gradients will be oriented perpendicular to the isosurfaces and can be located anywhere in

space. We will refer to the main random function—the scalar field itself—$\mathbf{Z}_\alpha$ simply as $\mathbf{Z}$, and its set of samples as $\mathbf{x}_\alpha$ while the second random function $\mathbf{Z}_\beta$—the gradient of the scalar field—will be referred to as $\partial\mathbf{Z}/\partial u$ and its samples as $\mathbf{x}_\beta$, so that we can capture the relationship between the potential field $\mathbf{Z}$ and its gradient as

$$\frac{\partial\mathbf{Z}}{\partial u}(x) = \lim_{p\to 0}\frac{\mathbf{Z}(x+pu)-\mathbf{Z}(x)}{p} \tag{1}$$

It is also important to keep the values of every individual synchronal layer identified since they have the same scalar field value. Therefore, samples that belong to a single layer $k$ will be expressed as a subset denoted using superscript as $\mathbf{x}_\alpha^k$ and every individual point by a subscript, $\mathbf{x}_{\alpha\,i}^k$ (see figure 2).

Note that in this context data does not have any meaningful physical parameter associated with it that we want to interpolate as long as stratigraphic deposition follows gradient direction. Therefore

the two constraints we want to conserve in the interpolated scalar field are: (i) all points belonging to a determined interface $\mathbf{x}_{\alpha\,i}^k$ must have the same scalar field value (i.e. there is an isosurface connecting all data points)

$$\mathbf{Z}(\mathbf{x}_{\alpha\,i}^k) - \mathbf{Z}(\mathbf{x}_{\alpha\,0}^k) = 0 \tag{2}$$

where $\mathbf{x}_{\alpha\,0}^k$ is a reference point of the interface and (ii) the scalar field will be perpendicular to the

poles $\mathbf{x}_\beta$ anywhere in 3-D space.

Considering equation 2, we do not care about the exact value at $\mathbf{Z}(\mathbf{x}_{\alpha\,i}^k)$ as long as it is constant at all points $\mathbf{x}_{\alpha\,i}^k$. Therefore, the random function $\mathbf{Z}$ in the CoKriging system (equation 4) can be substituted by equation 2. This formulation entails that the specific *scalar field values* will depend only on the gradients and hence at least one gradient is necessary to keep the system of equations

defined. The reason for this formulation rest on that by not fixing the values of each interface $\mathbf{Z}(\mathbf{x}_\alpha^k)$, the compression of layers—which is derived by the gradients—can propagate smoother beyond the given interfaces. Otherwise, the gradients will only have effect in the area within the boundaries of the two interfaces that contains the variable.

The algebraic dependency between $\mathbf{Z}$ and $\partial\mathbf{Z}/\partial u$ (equation 1) gives a mathematical definition of

the relation between the two variables avoiding the need of an empirical cross-variogram, enabling



instead the use of the derivation of the covariance function. This dependency must be taken into consideration in the computation of the drift of the first moment as well having a different function for each of the variables

$$\lambda F_1 + \lambda F_2 = f_{10} \tag{3}$$

where $F_1$ is a the polynomial of degree $n$ and $F_2$ its derivative. Having taken this into consideration, the resulting CoKriging system takes the form of:

$$
\begin{bmatrix}
\mathbf{C}_{\partial\mathbf{Z}/\partial\mathbf{u},\partial\mathbf{Z}/\partial\mathbf{v}} & \mathbf{C}_{\partial\mathbf{Z}/\partial\mathbf{u},\mathbf{Z}} & \mathbf{U}_{\partial\mathbf{Z}/\partial\mathbf{u}} \\
\mathbf{C}_{\mathbf{Z},\partial\mathbf{Z}/\partial\mathbf{u}} & \mathbf{C}_{\mathbf{Z},\mathbf{Z}} & \mathbf{U}_{\mathbf{Z}} \\
\mathbf{U}'_{\partial\mathbf{Z}/\partial\mathbf{u}} & \mathbf{U}'_{\mathbf{Z}} & \mathbf{0}
\end{bmatrix}
\begin{bmatrix}
\lambda_{\partial\mathbf{Z}/\partial u,\partial\mathbf{Z}/\partial v} & \lambda_{\partial\mathbf{Z}/\partial u, Z} \\
\lambda_{Z,\partial\mathbf{Z}/\partial u} & \lambda_{\mathbf{Z},\mathbf{Z}} \\
\mu_{\partial_u} & \mu_u
\end{bmatrix}
=
\begin{bmatrix}
\mathbf{c}_{\partial\mathbf{Z}/\partial\mathbf{u},\partial\mathbf{Z}/\partial\mathbf{v}} & \mathbf{c}_{\partial\mathbf{Z}/\partial\mathbf{u},\mathbf{Z}} \\
\mathbf{c}_{\mathbf{Z},\partial\mathbf{Z}/\partial\mathbf{u}} & \mathbf{c}_{\mathbf{Z},\mathbf{Z}} \\
\mathbf{f_{10}} & \mathbf{f_{20}}
\end{bmatrix}
\tag{4}
$$

where, $\mathbf{C}_{\partial\mathbf{Z}/\partial\mathbf{u}}$ is the gradient covariance-matrix; $\mathbf{C}_{\mathbf{Z},\mathbf{Z}}$ the covariance-matrix of the differences between each interface points to reference points in each layer

$$C_{\mathbf{x}^r_{\alpha\,i},\mathbf{x}^s_{\alpha\,j}} = C_{x^r_{\alpha,\,i}\,x^s_{\alpha,\,j}} - C_{x^r_{\alpha,\,0}\,x^s_{\alpha,\,j}} - C_{x^r_{\alpha,\,i}\,x^s_{\alpha,\,0}} + C_{x^r_{\alpha,\,0}\,x^s_{\alpha,\,0}} \tag{5}$$

(see Appendix B2 for further analysis); $\mathbf{C}_{\mathbf{Z},\partial\mathbf{Z}/\partial\mathbf{u}}$ encapsulates the cross-covariance function; and $\mathbf{U}_{\mathbf{Z}}$ and $\mathbf{U}'_{\partial\mathbf{Z}/\partial\mathbf{u}}$ are the drift functions and their gradient, respectively. On the right hand side we find the vector of the matrix system of equations, being $\mathbf{c}_{\partial\mathbf{Z}/\partial\mathbf{u},\partial\mathbf{Z}/\partial\mathbf{v}}$ the gradient of the covariance function to the point $\mathbf{x}$ of interest; $\mathbf{c}_{\mathbf{Z},\partial\mathbf{Z}/\partial\mathbf{u}}$ the cross-covariance; $\mathbf{c}_{\mathbf{Z},\mathbf{Z}}$ the actual covariance func-

tion; and $\mathbf{f_{10}}$ and $\mathbf{f_{20}}$ the gradient of the drift functions and the drift functions themselves. Lastly, the unknown vectors are formed by the corresponding weights, $\lambda$, and constants of the drift functions, $\mu$. A more detail inspection of this system of equations is carried out in Appendix B.

As we can see in equation 4, it is possible to solve the Kriging system for the scalar field $\mathbf{Z}$ (second column in the weights vector), as well as its derivative $\partial\mathbf{Z}/\partial u$ (first column in the weights vector).

Even though the main goal is the segmentation of the layers, which is done using the value of $\mathbf{Z}$ (see Section 2.2.1), the gradient of the scalar field can be used for further mathematical applications, such as meshing, geophysical forward calculations or locating geological structures of interest (e.g. spill points of a hydrocarbon trap).

Furthermore, since the choice of covariance parameters is ad hoc (Appendix D show the used

covariance function in GemPy), the uncertainty derived by the Kriging interpolation does not bear any physical meaning. This fact promotes the idea of only using the mean value of the Kriging solution. For this reason it is recommended to solve the Kriging system (equation 4) in its dual form (Matheron, 1981, see Appendix C).





### 2.2 Geological model interpolation using *GemPy*

#### 2.2.1 From scalar field to geological block model

In most scenarios the goal of structural modeling is to define the spatial distribution of geological structures, such as layers interfaces and faults. In practice, this segmentation usually is done either by using a volumetric discretization or by depicting the interfaces as surfaces.

The result of the Kriging interpolation is the random function $\mathbf{Z}(x)$ (and its gradient $\partial \mathbf{Z}/\partial u(x)$,
which we will omit in the following), which allows the evaluation of the value of the scalar field at any given point $x$ in space. From this point on, the easiest way to segment the domains is to discretize the 3-D space (e.g. we use a regular grid in figure 3). First, we need to calculate the scalar value at every interface by computing $\mathbf{Z}(\mathbf{x}_{\alpha,i}^{k})$ for every interface $k_i$. Once we know the value of the scalar field at the interfaces, we evaluate every point of the mesh and compare their value to
those at the interfaces, identifying every point of the mesh with a topological volume. Each of these compartmentalizations will represent each individual domain, this is, each lithology of interest (see figure 3a).

At the time of this manuscript preparation, *GemPy* only provides rectilinear grids but it is important to notice that the computation of the scalar field happens in continuous space, and therefore
allows the use of any type of mesh. The result of this type of segmentation is referred in *GemPy* as a *lithology block*.

The second segmentation alternative consist on locating the layer isosurfaces. *GemPy* makes use of the marching cube algorithm (Lorensen and Cline, 1987) provided by the *scikit-image* library (van der Walt et al., 2014). The basics of the marching cube algorithm are quite intuitive: (i) First,
we discretize the volume in 3-D voxels and by comparison we look if the value of the isosurface we want to extract falls within the boundary of every single voxel; (ii) if so, for each edge of the voxel, we interpolate the values at the corners of the cube to obtain the coordinates of the intersection between the edges of the voxels and the isosurface of interest, commonly referred to as vertices; (iii) those intersections are analyzed and compared against all possible configurations to define the
simplices (i.e. the vertices which form an individual polygon) of the triangles. Once we obtain the coordinates of vertices and their correspondent simplices, we can use them for visualization (see Section 3.1) or any sub-sequential computation that may make use of them (e.g. weighted voxels). For more information on meshing algorithms refer to (Geuzaine and Remacle, 2009).

**Listing 1.** Code to generate a single scalar field model (as seen in figure 2) and plotting a section of a regular grid (figure 3a) and extracting surfaces points at the interfaces.

```
import gempy as gp

     # Main data management object containing
     geo_data = gp.create_data(extent=[0, 20, 0, 10, -10, 0],
                               resolution=[100, 10, 100],
path_o="paper_Foliations.csv",
```



```
                          path_i="paper_Points.csv")

        # Creating object with data prepared for interpolation and compiling
        interp_data = gp.InterpolatorData(geo_data)

        # Computing result
        lith, fault = gp.compute_model(interp_data)

        # Plotting result: scalar field
gp.plot_scalar_field(geo_data, lith[1], 5, plot_data=True)

        # Plotting result: lithology block
        gp.plot_section(geo_data, lith[0], 5, plot_data=True)

# Getting vertices and faces
        vertices, simpleces = gp.get_surfaces(interp_data, lith[1], [fault[1]], original_scale=True)
```




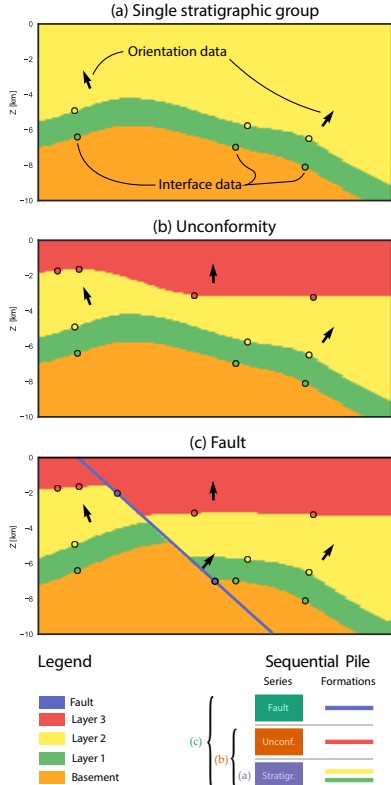

**Figure 3.** Example of different lithological units and their relation to scalar fields. a) Simple stratigraphic sequence generated from a scalar field as product of the interpolation of interface points and orientation gradients. b) Addition of an unconformity horizon from which the unconformity layer behaves independently from the older layers by overlying a second scalar field. c) Combination of unconformity and faulting using three scalar fields.

### 2.2.2 Combining scalar fields: Depositional series and faults

In reality, most geological settings are formed by a concatenation of depositional phases partitioned
by unconformity boundaries and subjected to tectonic stresses which displace and deform the layers.
While the interpolation is able to represent realistic folding—given enough data—the method fails
to describe discontinuities. To overcome this limitation, it is possible to combine several scalar fields
to recreate the desired result.

So far the implemented discontinuities in *GemPy* are unconformities and infinite faults. Both types
are computed by specific combinations of independent scalar fields. We call these independent scalar
fields *series* (from stratigraphic series in accordance to the use in GeoModeller 3-D Calcagno et al.,
2008) and in essence, they represent a subset of grouped interfaces—either layers or fault planes—





that are interpolated together and therefore their spatial location affect each other. To handle and visualize these relationships, we use a so called sequential pile; representing the order—from the first scalar field to the last—and the grouping of the layers (see figure 3).

Modeling unconformities is rather straightforward. Once we have grouped the layers into their respective series, younger series will overlay older ones beyond the unconformity. The scalar fields themselves, computed for each of these series, could be seen as a continuous depositional sequence in the absence of an unconformity.

**Listing 2.** Extension of the code in Listing 1 to generate an unconformity by using two scalar fields. The corresponding model is shown in figure 3b)

```python
import gempy as gp

# Main data management object containing
geo_data = gp.create_data(extent=[0, 20, 0, 10, -10, 0],
                          resolution=[100, 10, 100],
                          path_o="paper_Foliations.csv",
                          path_i="paper_Points.csv")

# Defining the series of the sequential pile
gp.set_series(geo_data,
              {'younger_serie' : 'Unconformity', 'older_serie': ('Layer1', 'Layer2')},
              order_formations= ['Unconformity', 'Layer2', 'Layer1'])

# Creating object with data prepared for interpolation and compiling
interp_data = gp.InterpolatorData(geo_data)

# Computing result
lith, fault = gp.compute_model(interp_data)

# Plotting result
gp.plot_section(geo_data, lith[0], 5, plot_data=True)
```

Faults are modeled by the inclusion of an extra drift term into the kriging system (Marechal, 1984):

$$
\begin{bmatrix}
\mathbf{C}_{\partial \mathbf{Z}/\partial \mathbf{u}, \partial \mathbf{Z}/\partial \mathbf{v}} & \mathbf{C}_{\partial \mathbf{Z}/\partial \mathbf{u}, \mathbf{Z}} & \mathbf{U}_{\partial \mathbf{Z}/\partial \mathbf{u}} & \mathbf{F}_{\partial \mathbf{Z}/\partial \mathbf{u}} \\
\mathbf{C}_{\mathbf{Z}, \partial \mathbf{Z}/\partial \mathbf{u}} & \mathbf{C}_{\mathbf{Z}, \mathbf{Z}} & \mathbf{U}_{\mathbf{Z}} & \mathbf{F}_{\mathbf{Z}} \\
\mathbf{U}'_{\partial \mathbf{Z}/\partial \mathbf{u}} & \mathbf{U}'_{\mathbf{Z}} & \mathbf{0} & \mathbf{0} \\
\mathbf{F}'_{\partial \mathbf{Z}/\partial \mathbf{u}} & \mathbf{F}'_{\mathbf{Z}} & \mathbf{0} & \mathbf{0}
\end{bmatrix}
\begin{bmatrix}
\lambda_{\partial \mathbf{Z}/\partial u, \partial \mathbf{Z}/\partial v} & \lambda_{\partial \mathbf{Z}/\partial u, Z} \\
\lambda_{Z, \partial \mathbf{Z}/\partial u} & \lambda_{\mathbf{Z}, \mathbf{Z}} \\
\mu_{\partial u} & \mu_{u} \\
\mu_{\partial f} & \mu_{f}
\end{bmatrix}
=
\begin{bmatrix}
\mathbf{c}_{\partial \mathbf{Z}/\partial u, \partial \mathbf{Z}/\partial v} & \mathbf{c}_{\partial \mathbf{Z}/\partial u, \mathbf{Z}} \\
\mathbf{c}_{\mathbf{Z}, \partial \mathbf{Z}/\partial u} & \mathbf{c}_{\mathbf{Z}, \mathbf{Z}} \\
\mathbf{f_{10}} & \mathbf{f_{20}} \\
\mathbf{f_{10}} & \mathbf{f_{20}}
\end{bmatrix}
\tag{6}
$$

which is a function of the faulting structure. This means that for every location $\mathbf{x}_0$ the drift function will take a value depending on the fault compartment—i.e. a segmented domain of the fault network—and other geometrical constrains such as spatial influence of a fault or variability of the offset. To obtain the offset effect of a fault, the value of the drift function has to be different at each of its sides. The level of complexity of the drift functions will determine the quality of the characterization as well as its robustness. Furthermore, finite or localize faults can be recreated by selecting an adequate function that describe those specific trends.





**Listing 3.** Code to generate an model with an unconformity and a fault using three scalar fields model (as seen in figure 3c) and the visualization 3D using vtk (see figure 5).

```
import gempy as gp

# Main data management object containing
geo_data = gp.create_data(extent=[0,20,0,10,-10,0],
                          resolution=[100,10,100],
                          path_o = "paper_Foliations.csv",
path_i = "paper_Points.csv")

# Defining the series of the sequential pile
gp.set_series(geo_data, series_distribution={'fault_serie1': 'fault1',
                        'younger_serie' : 'Unconformity',
'older_serie': ('Layer1', 'Layer2')},
order_formations= ['fault1', 'Unconformity', 'Layer2', 'Layer1'])

# Creating object with data prepared for interpolation and compiling
interp_data = gp.InterpolatorData(geo_data)

# Computing result
lith, fault = gp.compute_model(interp_data)

# Plotting result
gp.plot_section(geo_data, lith[0], 5, plot_data=True)

# Getting vertices and faces and pltting
vertices, simpleces = gp.get_surfaces(interp_data,lith[1], [fault[1]], original_scale=True)
gp.plot_surfaces_3D(geo_data, ver_s, sim_s)
```

The computation of the segmentation of fault compartments (called *fault block* in *GemPy*)—prior to the inclusion of the fault drift functions which depends on this segmentation—can be performed with the potential-field method itself. In the case of multiple faults, individual drift functions have to be included in the kriging system for each fault, representing the subdivision of space that they

produce. Naturally, younger faults may offset older tectonic events. This behavoir is replicated by recursively adding drift functions of younger faults to the computation of the older *fault blocks*. To date, the fault relations—i.e. which faults offset others—is described by the user in a boolean matrix. An easy to use implementation to generate fault networks is being worked on at the time of manuscript preparation.

An important detail to consider is that drift functions will bend the isosurfaces according to the given rules but they will conserve their continuity. This differs from the intuitive idea of offset, where the interface presents a sharp jump. This fact has direct impact in the geometry of the final model, and can, for example, affect certain meshing algorithms. Furthermore, in the ideal case of choosing the perfect drift function, the isosurface would bend exactly along the faulting plane. At

the current state *GemPy* only includes the addition of an arbitrary integer to each segmented volume. This limits the quality to a constant offset, decreasing the sharpness of the offset as data deviates from that constrain. Any deviation from this theoretical concept, results in a bending of the layers as



they approximate the fault plane to accommodate to the data, potentially leading to overly smooth transitions around the discontinuity.

### 2.3 "Under the hood": The *GemPy* architecture

#### 2.3.1 The graph structure

The architecture of *GemPy* follows the Python Software Foundation recommendations of modularity and reusability (van Rossum et al., 2001). The aim is to divide all functionality into independent small logical units in order to avoid duplication, facilitate readability and make changes to the code base easier.

The design of *GemPy* revolves around an automatic differentiation (AD) scheme. The main constraint is that the mathematical functions need to be continuous from the input parameters (in probabilistic jargon priors) to the cost function (or likelihoods), and therefore the code must be written in the same language (or at the very least compatible) to automatically compute the derivatives. In practice, this entails that any operation involved in the AD must be coded symbolically using the library *Theano* (see Section 2.3.2 for further details). One of the constrains of writing symbolically is the a priori declaration of the possible input parameters of the graph which will behave as latent variables—i.e. the parameters we try to tune for optimization or uncertainty quantification—while leaving others involved parameters constant either due to their nature or because of the relative slight impact of their variability. This rigidity dictates the whole design of input data management that needs to revolved around the preexistent symbolic graph.

*GemPy* encapsulates this creation of the symbolic graph in its the module `theanograph`. Due to the significant complexity to program symbolically, features shipped in *GemPy* that rely heavily in external libraries are not written in *Theano*. The current functionality written in *Theano* can be seen in the figure 4 and essentially it encompasses all the interpolation of the geological modeling (section 2.1) as well as forward calculation of the gravity (section 3.2).

Regarding data structure, we make use of the Python package *pandas* (McKinney, 2011) to store and prepare the input data for the symbolic graph (red nodes in figure 4) or other processes, such as visualization. All the methodology to create, export and manipulate the original data is encapsulated in the class `DataManagement`. This class has several child classes to facilitate specific precomputation manipulations of data structures (e.g. for meshing). The aim is to have all constant data prepared before any inference or optimization is carried out to minimize the computational overhead of each iteration as much as possible.

It is important to keep in mind that, in this structure, once data enters the part of the symbolic graph, only algebraic operations are allowed. This limits the use of many high-level coding structures (e.g. dictionaries or undefined loops) and external dependencies. As a result of that, the preparation of data must be exhaustive before starting the computation. This includes ordering the data within





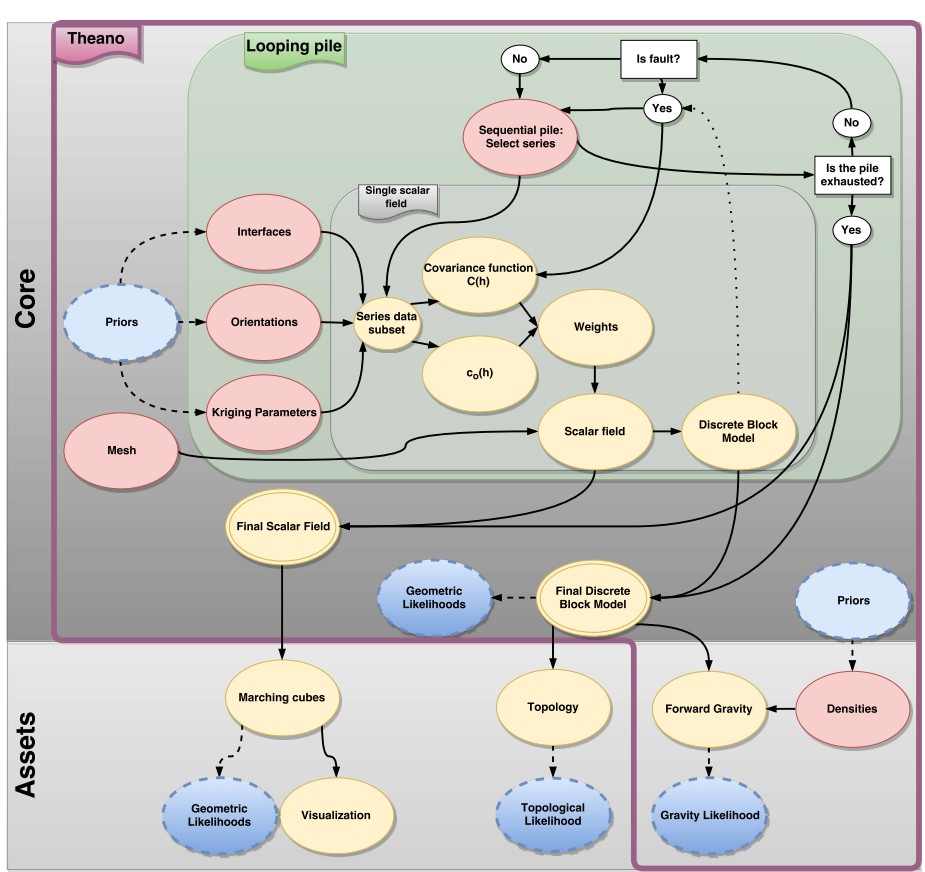

**Figure 4.** Graph of the logical structure of *GemPy* logic. There are several levels of abstraction represented. (i) The first division is between the implicit interpolation of the geological modeling (dark gray) and other subsequent operations for different objectives (light gray). (ii) All the logic required to perform automatic differentiation is presented under the "Theano" label (in purple) (iii) The parts under labels "Looping pile" (green) and "Single potential field" (gray), divide the logic to control the input data of each necessary scalar field and the operations within one of them. (iv) Finally, each superset of parameters is color coded according to their probabilistic nature and behavior in the graph: in blue, stochastic variables (priors or likelihoods); in yellow, deterministic functions; and in red the inputs of the graph, which are either stochastic or constant depending on the problem.



the arrays, passing the exact lengths of the subsets we will need later on during the interpolation or the calculation of many necessary constant parameters. The preprocessing of data is done within the
sub-classes of `DataManagement`, the `InterpolatorData` class–of which an instance is used to call the *Theano* graph—and `InterpolatorClass`—which creates the the *Theano* variables and compiles the symbolic graph.

The rest of the package is formed by—an always growing—series of modules that perform different tasks using the geological model as input (see Section 3 and the assets-area in figure 4).

### 2.3.2   Theano

Efficiently solving a large number of algebraic equations, and especially their derivatives, can easily get unmanageable in terms of both time and memory. Up to this point we have referenced many times *Theano* and its related terms such as automatic differentiation or symbolic programming. In this section we will motivate its use and why its capabilities make all the difference in making
implicit geological modeling available for uncertainty analysis.

*Theano* is a Python package that takes over many of the optimization tasks in order to create a computationally feasible code implementation. *Theano* relies on the creation of symbolical graphs that represent the mathematical expressions to compute. Most of the extended programming paradigms (e.g. procedural languages and object-oriented programming; see Normark, 2013) are executed se-
quentially without any interaction with the subsequent instructions. In other words, a later instruction has access to the memory states but is clueless about the previous instructions that have modified mentioned states. In contrast, symbolic programming define from the beginning to the end not only the primary data structure but also the complete logic of a function , which in turn enables the optimization (e.g. redundancy) and manipulation (e.g. derivatives) of its logic.

Within the Python implementation, *Theano* create an acyclic network graph where the parameters are represented by nodes, while the connections determine mathematical operators that relate them. The creation of the graph is done in the class `theanograph`. Each individual method corresponds to a piece of the graph starting from the input data all the way to the geological model or the forward gravity (see figure 4, purple Theano area).

The symbolic graph is later analyzed to perform the optimization, the symbolic differentiation and the compilation to a faster language than Python (C or CUDA). This process is computational demanding and therefore it must be avoided as much as possible.

Among the most outstanding optimizers shipped with *Theano* (for a detailed description see Theano Development Team, 2016), we can find : (i) the canonicalization of the operations to re-
duce the number of duplicated computations, (ii) specialization of operations to improve consecutive element-wise operations, (iii) in-place operations to avoid duplications of memory or (iv) Open MP parallelization for CPU computations. These optimizations and more can speed up the code an order of magnitude.



However, although *Theano* code optimization is useful, the real game-changer is its capability
to perform automatic differentiation. There is extensive literature explaining all the ins and outs
and intuitions of the method since it is a core algorithm to train neural networks (e.g. a detailed
explanation is given by Baydin et al., 2015). Here, we will highlight the main differences with
numerical approaches and how they can be used to improve the modeling process.

Many of the most advanced algorithms in computer science rely on an inverse framework i.e. the
result of a forward computation $f(\mathbf{x})$ influences the value of one or many of the $\mathbf{x}$ latent variables (e.g.
neuronal networks, optimizations, inferences). The most emblematic example of this is the optimiza-
tion of a cost function. All these problems can be described as an exploration of a multidimensional
manifold $f : \mathbb{R}^N \to \mathbb{R}$. Hence the gradient of the function $\nabla f = \left( \frac{\partial f}{\partial x_1}, \frac{\partial f}{\partial x_2}, \ldots, \frac{\partial f}{\partial x_n} \right)$ becomes key
for an efficient analysis. In case that the output is also multidimensional—i.e. $f : \mathbb{R}^N \to \mathbb{R}^M$—the
entire manifold gradient can be expressed by the Jacobian matrix

$$
Jf = \begin{bmatrix} \frac{\partial f_1}{\partial x_1} & \cdots & \frac{\partial f_1}{\partial x_n} \\ \vdots & \ddots & \vdots \\ \frac{\partial f_n}{\partial x_1} & \cdots & \frac{\partial f_m}{\partial x_n} \end{bmatrix} \tag{7}
$$

of dimension $N \cdot M$, where $N$ is the number of variables and $M$ the number of functions that de-
pend on those variables. Now the question is how we compute the Jacobian matrix in a consistent
and efficient manner. The most straightforward methodology consists in approximating the derivate
by numerical differentiation applying finite differences approximations, for example a forward FD
scheme:

$$
\frac{\partial f_i}{\partial x_i} = \lim_{h \to 0} \frac{f(x_i + h) - f(x_i)}{h} \tag{8}
$$

where $h$ is a discrete increment. The main advantage of numerical differentiation is that it only
computes $f$—evaluated for different values of $x$—which makes it very easy to implement it in any
available code. By contrast, a drawback is that for every element of the Jacobian we are introducing
an approximation error that eventually can lead to mathematical instabilities. But above all, the main
limitation is the need of $2 \cdot M \cdot N$ evaluations of the function $f$, which quickly becomes prohibitively
expensive to compute in high-dimensional problems.

The alternative is to create the symbolic differentiation of $f$. This encompasses decomposing $f$
into its primal operators and applying the chain rule to the correspondent transformation by follow-
ing the rules of differentiation to obtain $f'$. However, symbolic differentiation is not enough since
the application of the chain rule leads to exponentially large expressions of $f'$ in what is known
as "expression swell" (Cohen, 2003). Luckily, these large symbolic expressions have a high level
of redundancy in their terms. This allows to exploit this redundancy by storing the repeated inter-
mediate steps in memory and simply invoke them when necessary, instead of computing the whole
graph every time. This division of the program into sub-routines to store the intermediate results—
which are invoked several times—is called dynamic programming (Bellman, 2013). The simplified



symbolic differentiation graph is ultimately what is called automatic differentiation (Baydin et al., 2015). Additionally, in a multivariate/multi-objective case the benefits of using AD increase linearly as the difference between the number of parameters $N$ and the number of objective functions $M$ get larger. By applying the same principle of redundancy explained above—this time between intermediate steps shared across multiple variables or multiple objective function—it is possible to reduce the number of evaluations necessary to compute the Jacobian either to $N$ in forward-propagation or to $M$ in back-propagation, plus a small overhead on the evaluations (for a more detailed description of the two modes of AD see Cohen, 2003).

*Theano* provides a direct implementation of the back-propagation algorithm, which means in practice that a new graph of similar size is generated per cost function (or, in the probabilistic inference, per likelihood function). Therefore, the computational time is independent of the number of input parameters, opening the door to solving high-dimensional problems.

## 3 ASSETS – Model analysis and further use

In this second half of the paper we will explore different features that complement and expand the construction of the geological model itself. These extensions are just some examples of how *GemPy* can be used as geological modeling engine for diverse research projects. The numerous libraries in the open-source ecosystem allow to choose the best narrow purpose tool for very specific tasks. Considering the visualization of *GemPy*, for instance: *matplotlib* (Hunter, 2007) for 2-D visualization, *vtk* for fast and interactive 3-D visualization, *steno3D* for sharing block models visualizations online— or even the open-source 3-D modeling software Blender (Blender Online Community, 2017) for creating high quality renderings and Virtual Reality, are only some examples of the flexibility that the combination of *GemPy* with other open-source packages offers. In the same fashion we can use the geological model as basis for the subsequent geophysical simulations and process simulations. Due to Python's modularity, combining distinct modules to extend the scope of a project to include the geological modeling process into a specific environment is effortless. In the next sections we will dive into some of the built-in functionality implemented to date on top of the geological modeling core. Current assets are: (i) 2-D and 3-D visualizations, (ii) forward calculation of gravity, (iii) topology analysis, (iv) uncertainty quantification (UQ) as well as (v) full Bayesian inference.

### 3.1 Visualization

The segmentation of meaningful units is the central task of geological modelling. It is often a prerequisite for engineering projects or process simulations. An intuitive 3-D visualization of a geological model is therefore a fundamntal requirement.

For its data and model visualization, *GemPy* makes use of freely available tools in the Python module ecosystem to allow the user to inspect data and modeling results from all possible angles. The





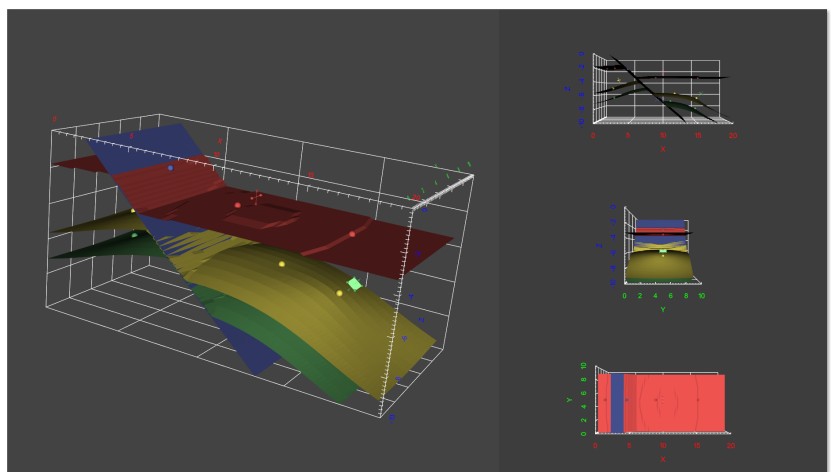

**Figure 5.** In-built *vtk* 3-D visualization of *GemPy* provides an interactive visualization of the geological model (left) and three additional orthogonal viewpoints (right) from different directions.

fundamental plotting library *matplotlib* (Hunter, 2007), enhanced by the statistical data visualization library *seaborn* (Waskom et al., 2017), provides the 2-D graphical interface to visualize input data and 2-D sections of scalar fields and geological models. In addition, making use of the capacities of

*pyqt* implemented with *matplotlib*, we can generate interactive sequence piles, where the user can not only visualize the temporal relation of the different unconformities and faulting events, but also modify it using intuitive drag and drop functionality (see figure 5).

On top of these features, *GemPy* offers in-built 3-D visualization based on the the open-source Visualization Toolkit (VTK Schroeder et al., 2004). It provides users with an interactive 3-D view

of the geological model, as well as three additional orthogonal viewpoints (see figure 5). The user can decide to plot just the data, the geological surfaces, or both. In addition to just visualizing the data in 3-D, *GemPy* makes use of the interaction capabilities provided by *vtk* to allow the user to move input data points on the fly via drag-and-drop. Combined with *GemPy*'s optimized modeling process (and the ability to use GPUs for efficient model calculation), this feature allows for data

modification with real-time updating of the geological model (in the order of milliseconds per scalar field). This functionality can not only improve the understanding of the model but can also help the user to obtain the desired outcome by working directly in 3-D space while getting direct visual feedback on the modeling results. Yet, due to the exponential increase of computational time with respect to the number of input data and the model resolution), very large and complex models may

have difficulties to render fast enough to perceive continuity on conventional computer systems.

For additional high quality visualization, we can generate vtk files using *pyevtk*. These files can later be loaded into external VTK viewer as Paraview (Ayachit, 2015) in order to take advantage of





its intuitive interface and powerful visualization options. Another natural compatibility exists with Blender (Blender Online Community, 2017) due to its use of Python as front-end. Using the Python

distribution shipped within a Blender installation, it is possible to import, run and automatically represent *GemPy*'s data and results (figure 1, see appendix F for code extension). This not only allow to render high quality images and videos but also to visualize the models in Virtual Reality, making use of the Blender Game engine and some of the plug-ins that enable this functionality.

    For sharing models, *GemPy* also includes functionality to upload discretized models to the Steno

3D platform (a freemium business model). Here, it is possible to visualize manipulate and shared the model with any number of people effortless by simple invitations or the distribution of a link.

    In short, *Gempy* is not limited to a unique visualization library. Currently *Gempy* gives support to many of the available visualization options to fulfill the different needs of the developers accordingly. However, these are not by all means the only possible alternatives and in the future we expect that

*GemPy* to be employed as backend of other further projects.

### 3.2   Gravity forward modeling

In recent years gravity measurements has increased in quality (Nabighian et al., 2005) and is by now a valuable additional geophysical data source to support geological modeling. There are different ways to include the new information into the modeling workflow, and one of the most common is

via inversions (Tarantola, 2005). Geophysics can validate the quality of the model in a probabilistic or optimization framework but also by back-propagating information, geophysics can improve automatically the modeling process itself. As a drawback, simulating forward geophysics adds a significant computational cost and increases the uncertainty to the parametrization of the model. However, due to the amount of uncorrelated information—often continuous in space—the inclusion

of geophysical data in the modeling process usually becomes significant to evaluate the quality of a given model.



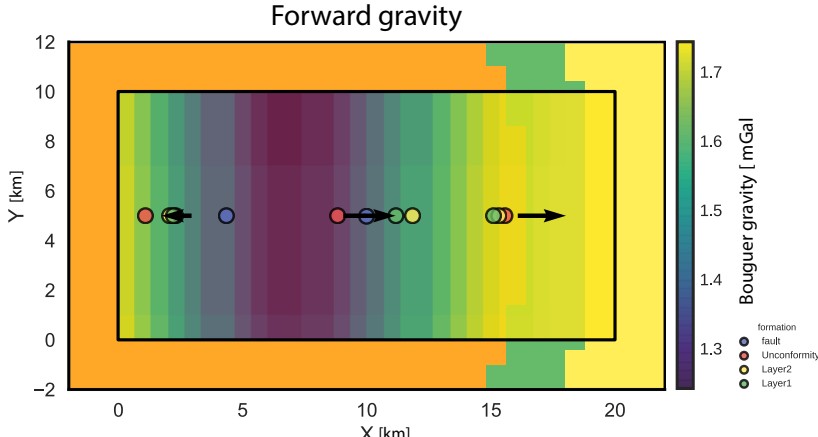

**Figure 6.** Forward gravity response overlayed on top of a XY cross section of the lithology block.

*GemPy* includes built-in functionality to compute forward gravity conserving the automatic differentiation of the package. It is calculated from the discretized block model applying the method of Nagy (1966) for rectangular prisms in the z direction,

$$F_z = G_\rho ||| x \ln(y+r) + y \ln(x+r) - z \arctan\left(\frac{xy}{zr}\right)|_{x_1}^{x_2}|_{y_1}^{y_2}|_{z_1}^{z_2} \qquad (9)$$

where $x$, $y$, and $z$ are the Cartesian components from the measuring point of the prism, $r$ the euclidean distance and $G_\rho$ the average gravity pull of the prism. This integration provides the gravitational pull of every voxel for a given density and distance in the component $z$. Taking advantage of the immutability of the involved parameters with the exception of density allow us to precompute the decomposition of $t_z$, leaving just its product with the weight $G_\rho$

$$F_z = G_\rho \cdot t_z \qquad (10)$$

as a recurrent operation.

As an example, we show here the forward gravity response of the geological model in figure 3c. The first important detail is the increased extent of the interpolated model to avoid boundary errors. In general, a padding equal to the maximum distance used to compute the forward gravity computation would be the ideal value. In this example (figure 6) we l add 10 km to the X and Y coordinates. The next step is to define the measurement 2-D grid—i.e. where to simulate the gravity response and the densities of each layers. The densities chosen are: 2.92, 3.1, 2.61 and 2.92 kg/m^3 for the basement, "Unconformity" layer (i.e. the layer on top of the unconformity), Layer 1 and Layer 2 respectively.

**Listing 4.** Computing forward gravity of a *GemPy* model for a given 2-D grid (see figure 6).

```
import matplotlib.pyplot as plt
import gempy as gp
```





```
# Main data management object containing. The extent must be large enough respect the forward
         gravity plane to account the effect of all cells at a given distance, $d$ to any spatial
         direction $x, y, z$.
     geo_data = gp.create_data(extent=[-10,30,-10,20,-10,0],
                               resolution=[50,50,50],
path_o = "paper_Foliations.csv",
                               path_i = "paper_Points.csv")

     # Defining the series of the sequential pile
     gp.set_series(geo_data, series_distribution={'fault_serie1': 'fault1',
'younger_serie' : 'Unconformity',
                                                   'older_serie': ('Layer1', 'Layer2')},
            order_formations= ['fault1', 'Unconformity', 'Layer2', 'Layer1'])

     # Creating object with data prepared for interpolation and compiling.
interp_data = gp.InterpolatorData(geo_data, output='gravity')

     # Setting the 2D grid of the airborn where we want to compute the forward gravity
     gp.set_geophysics_obj(interp_data_g,  ai_extent = [0, 20, 0, 10, -10, 0],
                           ai_resolution = [30,10])
     # Making all possible precomputations: Decomposing the value tz for every point of the 2D grid
         to each voxel
     gp.precomputations_gravity(interp_data_g, 25, densities=[2.92, 3.1, 2.61, 2.92])

# Computing gravity (Eq. 10)
     lith, fault, grav = gp.compute_model(interp_data_g, 'gravity')

     # Plotting lithology section
     gp.plot_section(geo_data, lith[0], 0, direction='z',plot_data=True)
     # Plotting forward gravity
     plt.imshow(grav.reshape(10,30), cmap='viridis', origin='lower', alpha=0.8, extent=[0,20,0,10])
```

The computation of forward gravity is a required step towards a fully coupled gravity inversion.
Embedding this step into a Bayesian inference allows to condition the initial data used to create the
model to the final gravity response. This idea will be further developed in Section 3.4.2.

### 3.3 Topology

The concept of topology provides a useful tool to describe adjacency relations in geomodels, such
as stratigraphic contacts or across-fault connectivity (for a more detailed introduction see Thiele
et al., 2016a, b). *GemPy* has in-built functionality to analyze the adjacency topology of its generated
models as Region Adjacency Graphs (RAGs), using the `topology_compute` method (see Listing
6). It can be directly visualized on top of model sections (see figure 7), where each unique topological
region in the geomodel is represented by a graph node, and each connection as a graph edge. The
function outputs the graph object G, the region centroid coordinates, a list of all the unique node
labels, and two look-up tables to conveniently reference node labels and lithologies





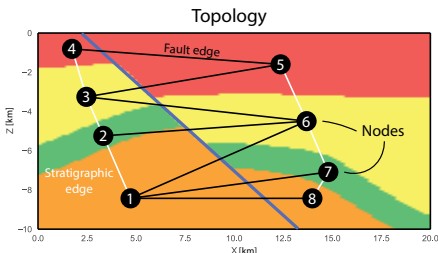

**Figure 7.** Section of the example geomodel with overlaid topology graph. The geomodel contains eight unique regions (graph nodes) and 13 unique connections (graph edges). White edges represent stratigraphic and un-conformity connections, while black edges correspond to across-fault connections.

To analyze the model topology, *GemPy* makes use of a general connected component labeling (CCL) algorithm to uniquely label all separated geological entities in 3-D geomodels. The algorithm is provided via the widely used, open-source, Python-based image processing library *scikit-image* (Van der Walt et al., 2014) by the function `skimage.measure.label`, which is based on the

optimized algorithms of (Fiorio and Gustedt, 1996; Wu et al., 2005). But just using CCL on a 3-D geomodel fails to discriminate a layer cut by a fault into two unique regions because in practice both sides of a fault are represented by the same label. To achieve the detection of edges across the fault, we need to precondition the 3-D geomodel matrix, which contains just the lithology information (layer id), with a 3-D matrix containing the information about the faults (fault block id).

This yields a 3-D matrix which combines the lithology information and the fault block information. This matrix can then be successfully labeled using CCL with a 2-connectivity stamp, resulting in a new matrix of uniquely labeled regions for the geomodel. From these, an adjacency graph is generated using `skimage.future.graph.RAG`, which created a Region Adjacency Graph (RAG) of all unique regions contained in a 2-D or 3-D matrix, representing each region with a node and

their adjacency relations as edges, successfully capturing the topology information of our geomodel. The connections (edges) are then further classified into either stratigraphic or across-fault edges, to provide further information. If the argument `compute_areas=True` was given, the contact area for the two regions of an edge is automatically calculated (number of voxels) and stored inside the adjacency graph.

**Listing 5.** Topology analysis of a GemPy geomodel.

```
...
Add Listing 3
...

# Computing result
lith, fault = gp.compute_model(interp_data)

# Compute topology
G, centroids, labels_unique, labels_lot, lith_lot = gp.topology_compute(geo_data, lith[0],
    fault[0], compute_areas=True)
```



```
# Plotting topology network
gp.plot_section(geo_data, lith[0], 5)
gp.topology_plot(geo_data, G, centroids)
```

### 3.4 Stochastic Geomodeling and probabilistic programming

Raw geological data is noisy and measurements are usually sparse. As a result, geological models contain significant uncertainties (Wellmann et al., 2010; Bardossy and Fodor, 2004; Lark et al., 2013; Caers, 2011; McLane et al., 2008; Chatfield, 1995) that must be addressed thoughtfully to reach a
plausible level of confidence in the model. However, treating geological modeling stochastically implies many considerations: (i) from tens or hundreds of variables involved in the mathematical equations which ones should be latent?; (ii) can we filter all the possible outcomes which represent unreasonable geological settings? and (iii) how can we use other sources of data—especially geophysics—to improve the accuracy of the inference itself?

The answers to these questions are still actively debated in research and are highly dependent on the type of mathematical and computational framework chosen. In the interpolation method explained in this paper, the parameters suitable to behave as a latent variables (see figure 4 for an overview of possible stochastic parameters) could be the interface points $\mathbf{x}_\alpha$ (i.e. the 3 Cartesian coordinates $x$, $y$, $z$), orientations $\mathbf{x}_\beta$ (i.e. the 3 Cartesian coordinates $x$, $y$, $z$ and the plane orientation normal $Gx$, $Gy$, $Gz$) or densities for the computation of the forward gravity. But not only parame-
ters with physical meaning are suitable to be considered stochastic. Many mathematical parameters used in the kriging interpolation—such as: covariance at distance zero $C_0$ (i.e. nugget effect) or the range of the covariance $r$ (see Appendix D for an example of a covariance function)—play a crucial role during the computation of the final models and, at best, are inferred by an educated guess to a
greater or lesser extent (Chiles et al., 2004; Calcagno et al., 2008). To tackle this problem in a strict manner, it would be necessary to combine Bayesian statistics, information theory and sensitivity analysis among other expertises, but in essence all these methodologies begin with a probabilistic programming framework.

*GemPy* is fully designed to be coupled with probabilistic frameworks, in particular with *pymc3*
(Salvatier et al., 2016) as both libraries are based on *Theano*.

*pymc* is a series of Python libraries that provide intuitive tools to build and subsequently infer complex probabilistic graphical models (see Koller and Friedman, 2009, and figure 13 as an example of a PGM). These libraries offer expressive and clean syntax to write and use statistical distributions and different samplers. At the moment two main libraries coexist due to their different strengths
and weaknesses. On the one hand, we have *pymc2* (Patil et al., 2010) written in FORTRAN and Python. *pymc2* does not allow gradient based sampling methods, since it does not have automatic differentiation capabilities. However, for that same reason, the model construction and debugging is more accessible. Furthermore, not computing gradients enables an easy integration with 3rd party



libraries and easy extensibility to other scientific libraries and languages. Therefore, for prototyp-
ing and lower dimensionality problems—where the posterior can be tracked by Metropolis-Hasting
methods (Haario et al., 2001)–*pymc2* is still the go-to choice.

On the other hand the latest version, *pymc3* (Salvatier et al., 2016), allows the use of next gen-
eration gradient-based samplers such as No-U-Turn Sampler (Hoffman and Gelman, 2014) or Au-
tomatic Variational Inference (Kucukelbir et al., 2015). These sampling methods are proving to be
a powerful tool to deal with multidimensional problems—i.e. models with high number of uncer-
tain parameters (Betancourt et al., 2017). The weakness of these methods are that they rely on the
computation of gradients, which in many cases cannot be manually derived. To circumvent this limi-
tation *pymc3* makes use of the AD capabilities of *Theano*. Being built on top of *Theano* confer to the
Bayesian inference process all the capabilities discussed in section 2.3.2 in exchange for the clarity
and flexibility that pure Python provides.

In this context, the purpose of *GemPy* is to fill the gap of complex algebra between the prior data
and observations, such as geophysical responses (e.g. gravity or seismic inversions) or geological
interpretations (e.g. tectonics, model topologies). Since *GemPy* is built on top of *Theano* as well, the
compatibility with both libraries is relatively straightforward. However, being able to encode most of
the conceivable probabilistic graphical models derived from, often, diverse and heterogeneous data
would be an herculean task. For this reason most of the construction of the PGM has to be coded by
the user using the building blocks that the *pymc* packages offer (see listing 6). By doing so, we can
guarantee full flexibility and adaptability to the necessities of every individual geological setting.

For this paper we will use *pymc2* for its higher readability and simplicity. *pymc3* architecture is
analogous with the major difference that the PGM is constructed in *Theano*—and therefore symbol-
ically (for examples using *pymc3* and *GemPy* check the online documentation detailed in Appendix
A2).

### 3.4.1   Uncertainty Quantification

An essential aspect of probabilistic programming is the inherent capability to quantify uncertainty.
Monte Carlo error propagation (Ogilvie, 1984) has been introduced in the field of geological model-
ing a few years ago (Wellmann et al., 2010; Jessell et al., 2010; Lindsay et al., 2012), exploiting the
automation of the model construction that implicit algorithms offer.

In this paper example (figure 8-Priors), we fit a normal distribution of standard deviation $300\,[\mathrm{m}]$
around the Z axis of the interface points in initial model (figure 3 c). In other words, we allows
to the interface points that define the model to oscillate independently along the axis Z accordingly
randomly—using normal distributions—and subsequently we compute the geomodels that these new
data describe.

The first step to the creation of a PGM is to define the parameters that are supposed to be stochas-
tic and the probability functions that describe them. To do so, *pymc2* provides a large selection of





distributions as well as a clear framework to create custom ones. Once we created the stochastic parameters we need to substitute the initial value in the *GemPy* database (`interp_data` in the snippets) for the corresponding *pymc2* objects. Next, we just need to follow the usual *GemPy* construction process—i.e. calling the `compute_model` function—wrapping it using a deterministic *pymc2* decorator to describe that these function is part of the probabilistic model (figure **??**). After creating the graphical model we can sample from the stochastic parameters using Monte Carlo sampling using *pymc2* methods.

**Listing 6.** Probabilistic model construction and inference using *pymc2* and *GemPy*: Monte Carlo forward simulation (see figure 8-Priors for the results).

```
...
Add Listing 3
...

# Coping the initial data
geo_data_stoch_init = deepcopy(interp_data.geo_data_res)
# MODEL CONSTRUCTION
# ==================
# Positions (rows) of the data we want to make stochastic
ids = range(2,12)

# List with the stochastic parameters. pymc.Normal attributes: Name, mean, std
interface_Z_modifier = [pymc.Normal("interface_Z_mod_"+str(i), 0., 1./0.01**2) for i in ids]

# Modifing the input data at each iteration
@pymc.deterministic(trace=True)
def input_data(value = 0, interface_Z_modifier = interface_Z_modifier,
               geo_data_stoch_init = geo_data_stoch_init,
               ids = ids, verbose=0):

    # First we extract from our original intep_data object the numerical data that
    # is necessary for the interpolation. geo_data_stoch is a pandas Dataframe
    geo_data_stoch = gp.get_data(geo_data_stoch_init, numeric=True)

    # Now we loop each id which share the same uncertainty variable. In this case, each layer.  We
    #   add the stochastic part to the initial value
        for num, i in enumerate(ids):
            interp_data.geo_data_res.interfaces.set_value(i, "Z",
                geo_data_stoch_init.interfaces.iloc[i]["Z"] + interface_Z_modifier[num])

    # Return the input data to be input into the modeling function. Due to the way pymc2
    # stores the traces we need to save the data as numpy arrays
        return interp_data.geo_data_res.interfaces[["X", "Y", "Z"]].values,
                interp_data.geo_data_res.orientations[["X", "Y", "Z", "dip", "azimuth",
                        "polarity"]].values

    # Computing the geological model
@pymc.deterministic(trace=True)
def gempy_model(value=0, input_data=input_data, verbose=False):

    # modify input data values accordingly
        interp_data.geo_data_res.interfaces[["X", "Y", "Z"]] = input_data[0]
```



```
# Gx, Gy, Gz are just used for visualization. The Theano function gets azimuth dip and
            polarity!!!
                interp_data.geo_data_res.orientations[["G_x", "G_y", "G_z", "X", "Y", "Z", 'dip',
                            'azimuth', 'polarity']] = input_data[1]
        # Some iterations will give a singular matrix, that's why we need to
        # create a try to not break the code.
            try:
                    lb, fb, grav = gp.compute_model(interp_data, outup='gravity')
return lb, fb, grav

            except np.linalg.linalg.LinAlgError as err:
        # If it fails (e.g. some input data combinations could lead to
        # a singular matrix and thus break the chain) return an empty model
# with same dimensions (just zeros)
                    if verbose:
                            print("Exception occured.")
                    return np.zeros_like(lith_block), np.zeros_like(fault_block), np.zeros_like(
                        grav_i)
        # Extract the vertices in every iteration by applying the marching cube algorithm
        @pymc.deterministic(trace=True)
        def gempy_surfaces(value=0, gempy_model=gempy_model):
                vert, simp = gp.get_surfaces(interp_data, gempy_model[0][1], gempy_model[1][1],
original_scale=True)
                return vert

        # We add all the pymc objects to a list
        params = [input_data, gempy_model, gempy_surfaces, *interface_Z_modifier]
        # We create the pymc model i.e. the probabilistic graph
        model = pymc.Model(params)
        runner = pymc.MCMC(model)

# BAYESIAN INFERENCE
        # ==================
        # Number of iterations
        iterations = 10000

# Inference. By default without likelihoods: Sampling from priors
        runner.sample(iter=iterations, verbose=1)
```

The suite of possible realization of the geological model are stored, as traces, in a database of choice (HDF5, SQL or Python pickles) for further analysis and visualization.

In 2-D we can display all possible locations of the interfaces on a cross-section at the center of the model (see figure 8-Priors-2-D representation), however the extension of uncertainty visualization to 3D is not as trivial. *GemPy* makes use of the latest developments in uncertainty visualization for 3-D structural geological modeling (e.g Lindsay et al., 2012, 2013a, b; Wellmann and Regenauer-Lieb, 2012). The first method consists on representing the probability of finding a given geological unit $F$



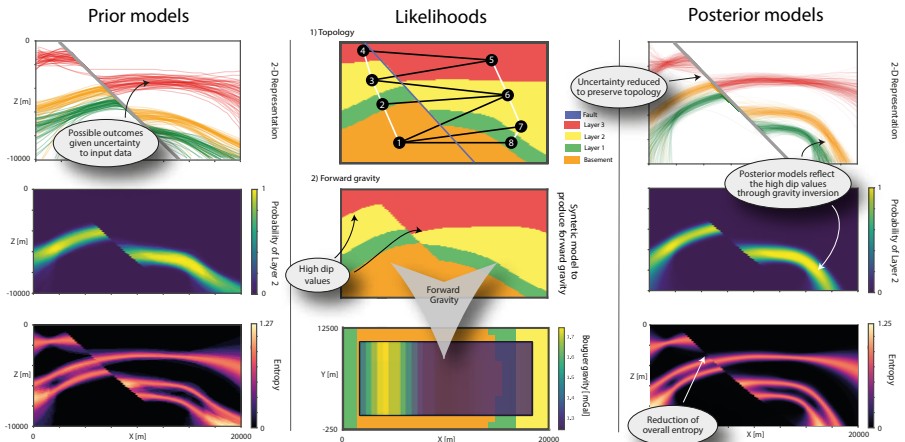

**Figure 8.** Probabilistic Programming results on a cross-section at the middle of the model ($Y = 10000\,[m]$). (i) Priors-UQ shows the uncertainty of geological models given stochastic values to the Z position of the input data ($\sigma = 300$): (top) 2-D interface representation ; (middle) probability of occurrence for Layer 1; (bottom) information entropy. (ii) Representation of data used as likelihood functions: (top) ideal topology graph; (middle) Synthetic model taken as reference for the gravity inversion; (bottom) Reference forward gravity overlain on top of an XY cross-section of the synthetic reference model. Posterior analysis after combining priors and likelihood in a Bayesian inference: (top) 2-D interface representation; (middle) probability of occurrence for Layer 1; (bottom) information entropy.

at each discrete location in the model domain. This can be done by defining a probability function

$$p_F(x) = \sum_{k \in n} \frac{I_{F_k}(x)}{n} \tag{11}$$

where n is the number of realizations and $I_{F_k}(x)$ is a indicator function of the mentioned geological unit (figure 8-Probability shows the probability of finding Layer 1). However this approach can only display each unit individually. A way to encapsulate geomodel uncertainty with a single pa-

rameter to quantify and visualize it, is by applying the concept of information entropy Wellmann and Regenauer-Lieb (2012), based on the general concept developed by (Shannon, 1948). For a discretized geomodel the information entropy $H$ (normalized by the total number of voxels $n$) can be defined as

$$H = -\sum_{i=1}^{n} p_i \log_2 p_i \tag{12}$$

where $p_F$ represents the probability of a layer at cell $x$. Therefore, we can use information entropy to compress our uncertainty into a single value at each voxel as an indication of uncertainty, reflecting the possible number of outcomes and their relative probability (see figure 8-Entropy).



### 3.4.2 Geological inversion: Gravity and Topology

Although computing the forward gravity has its own value for many applications, the main aim of
*GemPy* is to integrate all possible sources of information into a single probabilistic framework. The
use of likelihood functions in a Bayesian inference in opposition to simply rejection sampling has
been explored by the authors during the recent years (de la Varga and Wellmann, 2016; Wellmann
et al., 2017; Schaaf, 2017). This approach enables to tune the conditioning of possible stochastic
realizations by varying the probabilistic density function used as likelihoods. In addition, Bayesian
networks allow to combine several likelihood functions, generating a competition among the prior
distribution of the input data and likelihood functions resulting in posterior distributions that best
honor all the given information. To give a flavor of what is possible, we apply custom likelihoods to
the previous example based on, topology and gravity constrains in an inversion.

As, we have shown above, topological graphs can represent the connectivity among the segmented
areas of a geological model. As is expected, stochastic perturbations of the input data can rapidly
alter the configuration of mentioned graphs. In order to preserve a given topological configuration
partially or totally, we can construct specific likelihood functions. To exemplify the use of a topo-
logical likelihood function, we will use the topology computed in the section 3.3 derived from the
initial model realization (figure 7 or 8-Likelihoods) as "ideal topology". This can be based on an
expert interpretation of kinematic data or deduced from auxiliary data.

The first challenge is to find a metric that captures the similarity of two graphs. As a graph is noth-
ing but a set of nodes and their edges we can compare the intersection and union of two different sets
using the the Jaccard index (Jaccard, 1912; Thiele et al., 2016a). It calculates the ratio of intersection
and union of two given graphs A and B:

$$J(A,B) = \frac{A \cap B}{A \cup B} \tag{13}$$

The resulting ratio is zero for entirely different graphs, while the metric rises as the sets of edges
and nodes become more similar between two graphs and reaches exactly one for an identical match.
Therefore, the Jaccard index can be used to express the similarity of topology graphs as a single
number we can evaluate using a probability density function. The type of probability density function
used will determine the "strength" or likelihood that the mean graph represent. Here, we use a half
Cauchy distribution ($\alpha = 0$ and $\beta = 10^{-3}$) due to its tolerance to outliers.

**Listing 7.** Probabilistic model construction and inference using *pymc2* and *GemPy*: Bayesian Inference (see
figure 8 for the results).

```
...
Add Listing 6
...

# Computation of toplogy
@pymc.deterministic(trace=True)
def gempy_topo(value=0, gm=gempy_model, verbose=False):
```

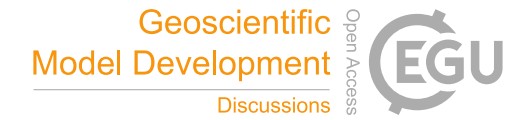



```
G, c, lu, lot1, lot2 = gp.topology_compute(geo_data, gm[0][0], gm[1], cell_number=0,
                direction="y")

             if verbose:
                  gp.plot_section(geo_data, gm[0][0], 0)
gp.topology_plot(geo_data, G, c)

             return G, c, lu, lot1, lot2

     # Computation of L2-Norm for the forward gravity
@pymc.deterministic
     def e_sq(value = original_grav, model_grav = gempy_model[2], verbose = 0):
             square_error =  np.sqrt(np.sum((value*10**-7 - (model_grav*10**-7))**2))
             return square_error

# Likelihoods
     # ==========
     @pymc.stochastic
     def like_topo_jaccard_cauchy(value=0, gempy_topo=gempy_topo, G=topo_G):
     """Compares the model output topology with a given topology graph G using an inverse Jaccard-
index embedded in a half-cauchy likelihood."""
     # jaccard-index comparison
             j = gp.Topology.compare_graphs(G, gempy_topo[0])
     # the last parameter adjusts the "strength" of the likelihood
             return pymc.half_cauchy_like(1 - j, 0, 0.001)
     @pymc.observed
     def inversion(value = 1, e_sq = e_sq):
             return pymc.half_cauchy_like(e_sq,0,0.1)

# We add all the pymc objects to a list
     params = [input_data, gempy_model, gempy_surfaces, gempy_topo, *interface_Z_modifier,
     like_topo_jaccard_cauchy, e_sq, inversion]

     # We create the pymc model i.e. the probabilistic graph
model = pymc.Model(params)
     runner = pymc.MCMC(model)

     # BAYESIAN INFERENCE
     # ===================
# Number of iterations
     iterations = 15000

     # Inference. Adaptive Metropolis
     runner.use_step_method(pymc.AdaptiveMetropolis, params, delay=1000)
runner.sample(iter = 20000, burn=1000, thin=20, tune_interval=1000, tune_throughout=True)
```

Gravity likelihoods exploit the spatial distribution of density which can be related to different lithotypes (Dentith and Mudge, 2014). To test the likelihood function based on gravity data, we first generate the synthetic "measured" data. This was done simply by computing the forward gravity
for one of the extreme models (to highlight the effect that a gravity likelihood can have) generated during the Monte Carlo error propagation in the previous section. This model is particularly characteristic by its high dip values (figure 8-Syntetic model to produce forward gravity). Once we have an





"observed" gravity, we can compare it to a simulated gravity response. To do so, we compare their
values applying an L2-norm encapsulating the difference into a single error value. This error value
acts as the input of the likelihood function, in this case, a half Cauchy ($\alpha = 0$ and $\beta = 10^{-1}$). This
probabilistic density function increases as we approach to 0 and at both extremes (very low or high
values of error) the function flatters to accommodate to possible measurement errors.

As sampler we use an adaptive Metropolis method (Haario et al., 2001, for a more in depth expla-
nation of samplers and their importance see de la Varga and Wellmann, 2016). This method varies the
metropolis sampling size according to the covariance function that gets updated every $n$ iterations.
For the results here exposed, we performed 20000 iterations, tuning the adaptive covariance every
1000 steps (a convergence analysis can be found in the Jupyter notebooks attached to the on-line
supplement of this paper).

As a result of applying likelihood functions we can appreciate a clear change in the posterior (i.e.
the possible outcomes) of the inference. A closer look shows two main zones of influence, each of
them related to one of the likelihood functions. On one hand, we observe a reduction of uncertainty
along the fault plane due to the restrictions that the topology function imposes by conditioning the
models to high Jaccard values. On the other hand, what in the first example—i.e. Monte Carlo error
propagation—was just an outlier, due to the influence of the gravity inversion, now it becomes the
norm bending the layers pronouncedly. In both cases, it is important to keep in mind that the grade of
impact into the final model is inversely proportional to the amount of uncertainty that each stochastic
parameter carries. Finally, we would like to remind the reader that the goal of this example is not
to obtain realistic geological models but to serve as an example how the in-built functionality of
*GemPy* can be used to handle similar cases.

## 965    4    Discussion

We have introduced *GemPy*, a Python library for implicit geomodelling with special emphasis on
the analysis of uncertainty. With the advent of powerful implicit methods to automate many of the
geological modeling steps, *GemPy* builds on these mathematical foundations to offer a reliable and
easy-to-use technology to generate complex models with only a few lines of code. In many cases—
and in research in particular—it is essential to have transparent software that allows full manipula-
tion and understanding of the logic beneath its front-end to honor the scientific method and allows
reproducibility by open-access to it.

Up until now, implicit geological modeling was limited to proprietary software suites—for the
petroleum industry (GoCad, Petrel, JewelSuite) or the mining sector (MicroMine, MIRA Geoscience,
GeoModeller, Leapfrog)—with an important focus on industry needs and user experience (e.g.
graphical user interfaces or data compatibilities). Despite the access to the APIs of many of these
softwares, their lack of transparency and the inability to fully manipulate any of the algorithms



represents a serious obstacle for conducting appropriate *reproducible* research. To overcome these limitations, many scientific communities—e.g. *simpeg* in geophysics (Cockett et al.), *astropy* in as-

tronomy (Robitaille et al., 2013) or *pynoddy* in kinematic structural modeling (Wellmann et al., 2016)—are moving towards the open-source frameworks necessary for the full application of the scientific method. In this regard, the advent of open-source programming languages such as R or Python are playing a crucial role in facilitating scientific programming and enabling the crucial re-producibility of simulations and script-based science. *GemPy* aims to fill the existing gap of implicit

modeling in the open-source ecosystem in geosciences that until now had to be filled by expensive general-purpose commercial softwares.

Implicit methods rely on interpolation functions to automate some or all the construction steps. Different mathematical approaches have been developed and improved in the recent years to tackle many of the challenges that particular geological settings pose (e.g. Lajaunie et al., 1997; Hillier

et al., 2014; Calcagno et al., 2008; Caumon et al., 2013; Caumon, 2010). A significant advantage of some of these methods is that they directly enable the re-computation of the entire structure when input data is changed or added. Furthermore, they can provide a geological meaningful interpola-tion function, for example considering deposition time (Caumon, 2010) or potential-fields (Lajaunie et al., 1997) to encapsulate the essence of geological deposition in different environments. The cre-

ation of *GemPy* has been made possible in a moment when the automation of geological modeling via implicit algorithms, as well as the maturity of the Python open-source ecosystem reached a point where a few thousand new lines of code are able to perform efficiently the millions of linear algebra operations and complex memory management necessary to create complex geological models. An important aspect in *GemPy*'s design has been the willingness to allow users to simply use *GemPy* as

a tool to construct geological models for different purposes as well as to encourage users to develop and expand *GemPy*'s code base itself. With the purpose to facilitate a low entry barrier we have taken two main structural decisions: (i) a clear separation between core features and extensible assets and (ii) combination of functional and object-oriented programming. The aim of this dual design is to give a friendly, easy-to-use front-end to the majority of users while keeping a modular structure of

the code for future contributors.

Using *GemPy* requires a minimum familiarity with the Python syntax. The lack of an advanced graphical interface to place the input data interactively forces the user to provide data sets with the coordinates and angular data. For this reason, for complex initial models *GemPy* could be seen more as a back-end library required to couple it with software providing 3-D graphical manipulation.

Due to the development team's background, *GemPy* is fully integrated with GeoModeller through the built-in library *pygeomod*. *GemPy* is able to read and modify GeoModeller projects directly, allowing to take advantage of their respective unique features. All input data of *GemPy* itself is kept in open, standard Python formats, making use of the flexible *pandas* DataFrames and powerful *numpy* arrays. Hence, every user will be able to freely manipulate the data at any given point.





*GemPy* has built-in functionality to visualize results using the main visualization libraries offered in Python: *matplotlib* for 2-D and *vtk* for 3-D and allows to export .vtk files for later visualization in common open-source tools for scientific visualizations such as ParaView. Altought *GemPy* does not include an evolved user interface, we offer certain level of interactivity using *GemPy*'s build-in 3-D visualization in vtk. Not only is the user able to move the input data via drag-and-drop, but

*GemPy* can immediately re-interpolate the perturbed model, enabling an extremely intuitive direct feedback on how the changes made affect the model. Visualization of vast model ensembles is also possible in 3-D using slider functionality. Future plans for the visualization of *GemPy* include virtual reality support to make data manipulation and model visualization more immersive and intuitive to use.

Another important feature of *GemPy* is the use of symbolic code. The lack of domain specific language allows the compilation of the code to a highly efficient language. Furthermore, since all the logic has to be described prior to the compilation, memory allocation and parallelization of the code can be optimized. *Theano* uses BLAS (Lawson et al., 1979) to perform the algebraic operations with out of the box Open MP (Dagum and Menon, 1998) capabilities for multi-core operations.

Additionally, parallel GPU computation is available and compatible with the use of CPUs, which allows to define certain operations to a specific device and even to split big arrays (e.g. grid) to multiple GPUs. In other words, the symbolic nature of the code enables the separation of the logic according to the individual advantages of each device—i.e. sequential computations to CPUs and parallel calculations to the GPUs—allowing for better use of the available hardware. Hence, this

scheme is portable to high performance computing in the same fashion.

Up to today, structural geological models have relied significantly on the best deterministic, explicit realization that an expert is capable to create using often noisy and sparse data. Research into the interpretation uncertainty of geological data sets (e.g. seismic data) has recognized the significant impact of interpreter education and bias on the extracted input data for geological models (e.g.

Bond et al., 2007; Bond, 2015). *GemPy*'s ability to be enveloped into probabilistic programming frameworks such as *pymc*, allows for the consideration of input data uncertainties and could provide a free, open-source foundation for developing probabilistic geomodeling workflows which integrate uncertainties from the very beginning of data interpretation, through the geomodel interpolation up to the geomodel application (e.g. flow simulations, economic estimations).

In the transition to a world dominated by data and optimization algorithms—e.g. deep neural networks or big data analytics—there are many attempts to apply those advances in geological modeling (Wang et al., 2017; Gonçalves et al., 2017). The biggest attempt to use data driven models in geology comes from geophysical inversions (Tarantola and Valette, 1982; Mosegaard and Tarantola, 1995; Sambridge and Mosegaard, 2002; Tarantola, 2005). Their approaches consist of using

the mismatch of one or many parameters, comparing model with reality, modifying them accordingly until reaching a given tolerance. However, since this solution is never unique, it is necessary

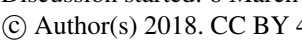


to enclose the space of possibilities by some other means. This prior approach to the final solution usually is made using polygonal or elliptic bodies leading to oversimplified geometry of the distinct lithological units. Other researchers use the additional data—geophysical data or other constrains (Jessell et al., 2010; Wellmann et al., 2014)— to validate multiple possible realizations of geological models generated either automatically by an interpolation function or manually. Here, the additional information is used as a hard deterministic filter of what is reasonable or not. The limitation of pure rejection filtering is that information does not propagate backward to modify the latent parameters that characterize the geological models what makes computational infeasible to explore high dimensional problems. In between these two approaches, we can find some attempts to reconcile both approaches meeting somewhere in the middle. An example of this is the approach followed in the software packages GeoModeller and SKUA. They optimize the layer densities, and when necessary the discretized model, to fit the geological model to the observed geophysical response. The consequence of only altering the discrete final model is that after optimization the original input data used for the construction of the geological model (i.e. interface points and orientation data) gets overwritten and consequently hard to reproduce.

We propose a more global approach. By embedding the geological model construction into a model-based machine learning framework (Bishop, 2013)—i.e. a Bayesian inference network. In short a Bayesian inference is a mathematical formulation to update beliefs in the light of new evidence. This statement applied to geological modeling is translated into keeping all or a subset of the parameters that generate the model uncertain and evaluate the quality of the model comparing its mismatch with additional data or geological knowledge encoded mathematically (de la Varga and Wellmann, 2016). In this way, we are able to utilize available information not only in a forward direction to construct models, but also propagate information backwards in an inverse scheme to refine the probabilistic distributions that characterize the modeling parameters. Compared with previous approaches, we do not only use the inversion to improve a deterministic model but instead to learn about the parameters that define the model to begin with. In recent years, we have shown how this approach may help closing the gap between geophysical inversions and geological modeling in an intuitive manner (Wellmann et al., 2017). At the end of the day, Bayesian inferences operate in a very similar way to how humans do: we create our best guess model; we compare it to the geophysical data or our geological knowledge and in case of disagreement we modify the input of the geological model in the direction we think is the best to honor the additional data.

Despite the convincing mathematical formulation of Bayesian inferences, there are caveats to be dealt with for practical applications. As mentioned in Section 3.4, the effective computational cost to perform such algorithms have prohibited its use beyond research and simplified models. However, recent developments in MCMC methods enable more efficient ways to explore the parametric space and hence opening the door to a significant increase on the complexity of geological models. An in-



depth study of the impact of gradient-based MCMC methods in geological modeling will be carried out in a future publication.

Nevertheless, performing AD does not come free of cost. The required code structures limit the use of libraries which do not perform AD themselves, which in essence imposes to rewrite most of the mathematical algorithms involved in the Bayesian network. Under these circumstances, we have rewritten in *Theano* the potential field method—with many of the add-ons developed in recent years (Calcagno et al., 2008)—and the computation of forward gravity responses for discrete rectangular

prisms.

Currently *GemPy* is in active development moving towards three core topics: (i) increasing the model based machine learning capabilities by exploiting gradient based methods and new types of likelihoods; (ii) post-processing of uncertainty quantification and its relation to decision theory and information theory; and (iii) exploring the new catalog of virtual reality and augmented reality solu-

tions to improve the visualization of both the final geological models and the building environment. However, ideally *GemPy* will function as a platform to create a vibrant open-source community to push forward geological modeling into the new machine learning era. Therefore, we hope to include functionality developed by other external users into the main package.

In conclusion, *GemPy* has evolved to a full approach for geological modelling in a probabilistic

programming framework. The lack of available open-source tools in geological modeling and the necessity of writing all the logic symbolically has pushed the project to an unexpected stand-alone size. However, this would not have been possible without the immense, ever-growing open-source community which provide numerous and high-quality libraries that enable the creation of powerful software with relative few new lines of code. And in the same fashion, we hope the community will

make use of our library to perform geological modeling transparently, reproducibly and incorporating the uncertainties inherent to earth sciences.

**Code availability**

GemPy is a free, open-source Python library licensed under the GNU Lesser General Public License v3.0 (GPLv3). It is hosted on the GitHub repository https://github.com/cgre-aachen/gempy (DOI:

10.5281/zenodo.1186118).

**Appendix A:  GemPy package information**

**A1   Installation**

Installing GemPy can be done in two ways: (i) Either by cloning the GitHub repository with `$ git clone https://github.com/cgre-aachen/gempy.git` and then manually installing it

by running the Python setup script in the repository: `$ python install.py` (ii) Or by using





the Python Package Index (PyPI) with the command `$ pip install gempy`, which directly downloads and installs the library.

### A2    Documentation

*GemPy*'s documentaion is hosted on `http://gempy.readthedocs.io/`, which provides a
general overview over the library and multiple in-depth tutorials. The tutorials are provided as Jupyter Notebooks, which provide the the convenient combination of documentation and executable script blocks in one document. The notebooks are part of the repository and located in the tutorials folder. See `http://jupyter.org/` for more information on installing and running Jupyter Notebooks.

### A3    Jupyter notebooks

We provide Jupyter notebooks as part of the online documentation. These notebooks can be executed in a local Python environment (if the required dependencies are correctly installed, see above). In addition, static versions of the notebooks can currently be inspected directly on the github repository web page or through the use of nbviewer. In addition, it is possible to run in-
teractive notebooks through the use of binder (provided through https://mybinder.org at the time of writing). For more details and up-to-date information, please refer to the repository page https://github.com/cgre-aachen/gempy.

### A4    Unit Tests

The *GemPy* package contains a set of tests, which can be executed in the standard Python testing
environment. If you cloned or downloaded the repository, then these tests can directly be performed by going to the package folder and run the pytest command: `$ pytest`

   If all tests are successful, you are ready to continue.

### Appendix B:  Kriging system expansion

The following equations have been derived from the work in Aug (2004); Lajaunie et al. (1997);
Chiles and Delfiner (2009).

### B1    Gradient Covariance-Matrix $\mathbf{C}_{\partial\mathbf{Z}/\partial\mathbf{u}}$

The gradient covariance-matrix, $\mathbf{C}_{\partial\mathbf{Z}/\partial\mathbf{u}}$, is made up of as many variables as gradient directions that are taken into consideration. In 3-D, we would have the Cartesian coordinates dimensions—$\mathbf{Z}/\partial x$, $\mathbf{Z}/\partial y$, and $\mathbf{Z}/\partial z$—and therefore, they will derive from the partial differentiation of the covariance
function $\sigma(x_i, x_j)$ of $\mathbf{Z}$.



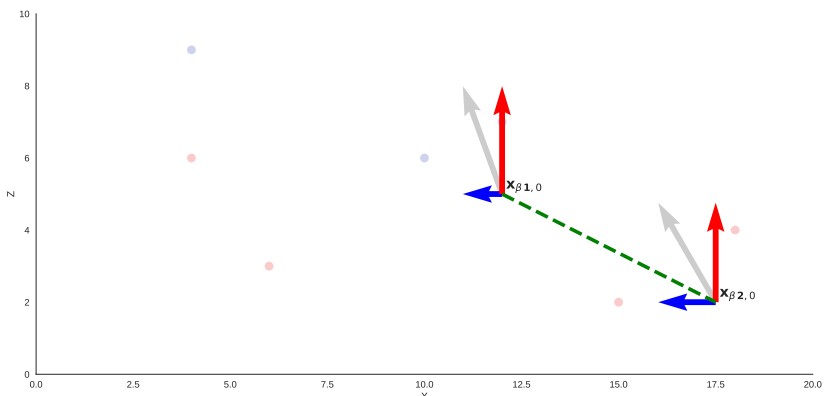

**Figure 9.** 2-D representation of the decomposition of the orientation vectors into Cartesian axis. Each Cartesian axis represent a variable of a sub CoKriging system. The dotted green line represent the covariance distance, $r$.

As in our case the directional derivatives used are the 3 Cartesian directions we can rewrite gradients covariance, $\mathbf{C}_{\partial \mathbf{Z}/\partial \mathbf{u}, \partial \mathbf{Z}/\partial \mathbf{v}}$ for our specific case as:

$$\mathbf{C}_{\partial \mathbf{Z}/\partial \mathbf{x}, \partial \mathbf{Z}/\partial \mathbf{y} \, \partial \mathbf{Z}/\partial \mathbf{z}} = \begin{bmatrix} \mathbf{C}_{\partial \mathbf{Z}/\partial \mathbf{x}, \partial \mathbf{Z}/\partial \mathbf{x}} & \mathbf{C}_{\partial \mathbf{Z}/\partial \mathbf{x}, \partial \mathbf{Z}/\partial \mathbf{y}} & \mathbf{C}_{\partial \mathbf{Z}/\partial \mathbf{x}, \partial \mathbf{Z}/\partial \mathbf{z}} \\ \mathbf{C}_{\partial \mathbf{Z}/\partial \mathbf{y}, \partial \mathbf{Z}/\partial \mathbf{x}} & \mathbf{C}_{\partial \mathbf{Z}/\partial \mathbf{y}, \partial \mathbf{Z}/\partial \mathbf{y}} & \mathbf{C}_{\partial \mathbf{Z}/\partial \mathbf{y}, \partial \mathbf{Z}/\partial \mathbf{z}} \\ \mathbf{C}_{\partial \mathbf{Z}/\partial \mathbf{z}, \partial \mathbf{Z}/\partial \mathbf{x}} & \mathbf{C}_{\partial \mathbf{Z}/\partial \mathbf{z}, \partial \mathbf{Z}/\partial \mathbf{y}} & \mathbf{C}_{\partial \mathbf{Z}/\partial \mathbf{z}, \partial \mathbf{Z}/\partial \mathbf{z}} \end{bmatrix} \tag{B1}$$

Notice, however, that covariance functions by definition are described in a polar coordinate system, and therefore it will be necessary to apply the chain rule for *directional derivatives*. Considering an isotropic and stationary covariance we can express the covariance function as:

$$\sigma(x_i, x_j) = C(r) \tag{B2}$$

with:

$$r = \sqrt{h_x^2 + h_y^2} \tag{B3}$$

therefore we need to apply the chain rule in partial differentiation. For the case of the covariance in a single direction:

$$\mathbf{C}_{\partial \mathbf{Z}/\partial \mathbf{u}, \partial \mathbf{Z}/\partial \mathbf{u}} = \frac{\partial^2 C_Z(r)}{\partial h_u^2} = \frac{\partial C_Z(r)}{\partial r} \frac{\partial}{\partial h_u} \left( \frac{\partial r}{\partial h_u} \right) + \frac{\partial}{\partial h_u} \left( \frac{\partial C_Z(r)}{\partial r} \right) \frac{\partial r}{\partial h_u} \tag{B4}$$

where:

$$\frac{\partial C_Z(r)}{\partial r} = \frac{\partial C_Z(r)}{\partial r} = C_Z'(r) \tag{B5}$$

$$\frac{\partial r}{\partial h_u} = \frac{h_u}{\sqrt{h_u^2 + h_v^2}} = -\frac{h_u}{r} \tag{B6}$$



$$\frac{\partial}{\partial h_u}\left(\frac{\partial r}{\partial h_u}\right) = \frac{\partial}{\partial h_u}\left(\frac{h_u}{\sqrt{h_u^2 + h_v^2}}\right) = -\frac{2h_u^2}{2\sqrt{h_u^2 + h_v^2}} + \frac{1}{\sqrt{h_u^2 + h_v^2}} = -\frac{h_u^2}{r^3} + \frac{1}{r} \tag{B7}$$

$$\frac{\partial}{\partial h_u}\left(\frac{\partial C_Z(r)}{\partial r}\right) = \frac{\partial C_Z'(r)}{\partial h_u} = \frac{\partial C_Z'(r)}{\partial r}\frac{\partial r}{\partial h_u} = -\frac{h_u}{r}C_Z'' \tag{B8}$$

Substituting:

$$\mathbf{C}_{\partial \mathbf{Z}/\partial \mathbf{u},\, \partial \mathbf{Z}/\partial \mathbf{u}} = C_Z'(r)\left(-\frac{h_u^2}{r^3} + \frac{1}{r}\right) - \frac{h_u}{r}C_Z''\frac{h_u}{r} = C_Z'(r)\left(-\frac{h_u^2}{r^3} + \frac{1}{r}\right) + \frac{h_u^2}{r^2}C_Z'' \tag{B9}$$

While in case of two different directions the covariance will be:

$$\mathbf{C}_{\partial \mathbf{Z}/\partial \mathbf{u},\, \partial \mathbf{Z}/\partial \mathbf{v}} = \frac{\partial^2 C_Z(r)}{\partial h_u h_v} = \frac{\partial C_Z(r)}{\partial r}\frac{\partial}{\partial h_v}\left(\frac{\partial r}{\partial h_u}\right) + \frac{\partial}{\partial h_v}\left(\frac{\partial C_Z(r)}{\partial r}\right)\frac{\partial r}{\partial h_u} \tag{B10}$$

with:

$$\frac{\partial}{\partial h_v}\left(\frac{\partial r}{\partial h_u}\right) = \frac{\partial}{\partial h_v}\left(\frac{h_u}{\sqrt{h_u^2 + h_v^2}}\right) = -\frac{h_u h_v}{r^3} \tag{B11}$$

$$\frac{\partial}{\partial h_v}\left(\frac{\partial C_Z(r)}{\partial r}\right) = \frac{\partial C_Z'(r)}{\partial h_v} = -C_Z''(r)\frac{h_v}{r} \tag{B12}$$

we have:

$$\mathbf{C}_{\partial \mathbf{Z}/\partial \mathbf{u},\, \partial \mathbf{Z}/\partial \mathbf{v}} = C_Z'(r)\left(-\frac{h_u h_v}{r^3}\right) + C_Z''(r)\frac{h_u h_v}{r^2} = \frac{h_u h_v}{r^2}\left(C_Z''(r) - \frac{C_Z'(r)}{r}\right) \tag{B13}$$

This derivation is independent to the covariance function choice, however, some covariances may lead to mathematical indeterminations.

### B2  Interface Covariance-Matrix

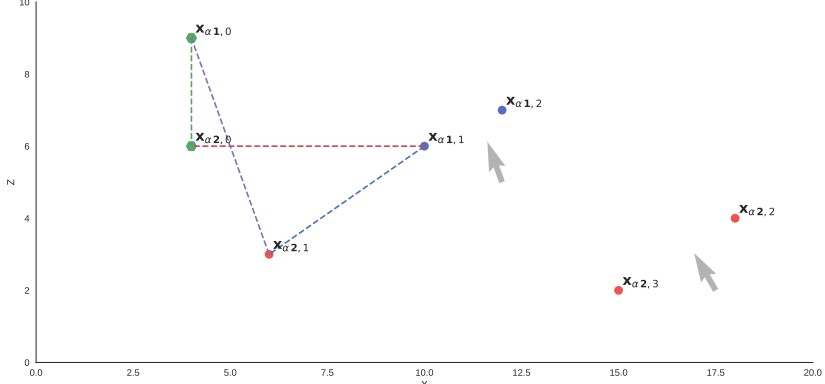

**Figure 10.** Distances $r$ involved in the computation of the interface subsystem of the interpolation. Because all covariances are relative to a reference point $x_{\alpha,0}^i$, all four covariances must be taken into account (equation B14)





In a practical sense, keeping the value of the scalar field at every interface unfixed forces us to consider the covariance between the points within an interface as well as the covariance between different layers following equation,

$$C_{\mathbf{x}^r_{\alpha\,i},\,\mathbf{x}^s_{\alpha\,j}} = C_{x^r_{\alpha,\,i}\,x^s_{\alpha,\,j}} - C_{x^r_{\alpha,\,0}\,x^s_{\alpha,\,j}} - C_{x^r_{\alpha,\,i}\,x^s_{\alpha,\,0}} + C_{x^r_{\alpha,\,0}\,x^s_{\alpha,\,0}} \tag{B14}$$

This lead to the subdivision of the CoKriging system respecting the interfaces:

$$\mathbf{C_{Z,\,Z}} = \begin{bmatrix} \mathbf{C_{\mathbf{x}^1_\alpha,\mathbf{x}^1_\alpha}} & \mathbf{C_{\mathbf{x}^1_\alpha,\mathbf{x}^2_\alpha}} & \cdots & \mathbf{C_{\mathbf{x}^1_\alpha,\mathbf{x}^s_\alpha}} \\ \mathbf{C_{\mathbf{x}^2_\alpha,\mathbf{x}^1_\alpha}} & \mathbf{C_{\mathbf{x}^2_\alpha,\mathbf{x}^2_\alpha}} & \cdots & \mathbf{C_{\mathbf{x}^2_\alpha,\mathbf{x}^s_\alpha}} \\ \vdots & \vdots & \ddots & \vdots \\ \mathbf{C_{\mathbf{x}^r_\alpha,\mathbf{x}^1_\alpha}} & \mathbf{C_{\mathbf{x}^r_\alpha,\mathbf{x}^2_\alpha}} & \cdots & \mathbf{C_{\mathbf{x}^r_\alpha,\mathbf{x}^s_\alpha}} \end{bmatrix} \tag{B15}$$

Combining Eq 5 and Eq B15 the covariance for the property *potential field* will look like:

$$\mathbf{C_{\mathbf{x}^r_\alpha,\mathbf{x}^s_\alpha}} = \begin{bmatrix} C_{x^1_1 x^1_1} - C_{x^1_0 x^1_1} - C_{x^1_1 x^1_0} + C_{x^1_0 x^1_0} & C_{x^1_1 x^1_2} - C_{x^1_0 x^1_2} - C_{x^1_1 x^1_0} + C_{x^1_0 x^1_0} & \cdots & C_{x^1_1 x^s_j} - C_{x^1_0 x^s_j} - C_{x^1_1 x^s_0} + C_{x^1_0 x^s_0} \\ C_{x^1_2 x^1_1} - C_{x^1_0 x^1_1} - C_{x^1_2 x^1_0} + C_{x^1_0 x^1_0} & C_{x^1_2 x^1_2} - C_{x^1_0 x^1_2} - C_{x^1_2 x^1_0} + C_{x^1_0 x^1_0} & \cdots & C_{x^1_2 x^s_j} - C_{x^1_0 x^s_j} - C_{x^1_j x^s_0} + C_{x^1_0 x^s_0} \\ \vdots & \vdots & \ddots & \vdots \\ C_{x^r_i x^s_1} - C_{x^r_0 x^s_1} - C_{x^r_i x^s_0} + C_{x^r_0 x^s_0} & C_{x^r_i x^s_2} - C_{x^r_0 x^s_2} - C_{x^r_i x^s_0} + C_{x^r_0 x^s_0} & \cdots & C_{x^r_i x^s_j} - C_{x^r_0 x^s_j} - C_{x^r_i x^s_0} + C_{x^r_0 x^s_0} \end{bmatrix} \tag{B16}$$

### B3 Cross-Covariance

In a CoKriging system, the relation between the interpolated parameters is given by a cross-covariance
function. As we saw above, the gradient covariance is subdivided into covariances with respect to the three Cartesian directions (Eq B1), while the interface covariance is detached from the covariances matrices with respect to each individual interface (Eq B15). In the same manner, the cross-covariance will reflect the relation of every interface to each gradient direction,

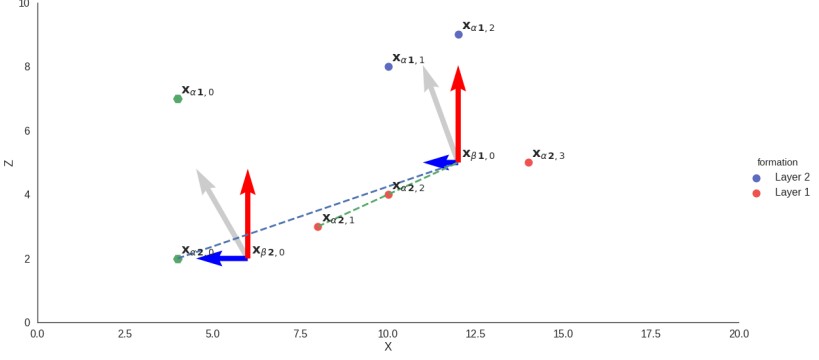

**Figure 11.** Distances $r$ involved in the computation of the cross-covariance function. In a similar fashion as before, all interface covariance are computed relative to a reference point in each layer $x^i_{\alpha,\,0}$





$$\mathbf{C}_{\mathbf{Z},\partial\mathbf{Z}/\partial\mathbf{u}} = \begin{bmatrix} \mathbf{C}_{\mathbf{x}_{\alpha\,1}^1,\partial\mathbf{Z}(\mathbf{x}_{\beta\,1})/\partial x} & \mathbf{C}_{\mathbf{x}_{\alpha\,2}^1,\partial\mathbf{Z}(\mathbf{x}_{\beta\,1})/\partial x} & \cdots & \mathbf{C}_{\mathbf{x}_{\alpha\,1}^1,\partial\mathbf{Z}(\mathbf{x}_{\beta\,1})/\partial y} & \cdots & \mathbf{C}_{\mathbf{x}_{\alpha\,i}^1,\partial\mathbf{Z}(\mathbf{x}_{\beta\,j})/\partial z} \\ \mathbf{C}_{\mathbf{x}_{\alpha\,1}^2,\partial\mathbf{Z}(\mathbf{x}_{\beta\,1})/\partial x} & \mathbf{C}_{\mathbf{x}_{\alpha\,2}^2,\partial\mathbf{Z}(\mathbf{x}_{\beta\,1})/\partial x} & \cdots & \mathbf{C}_{\mathbf{x}_{\alpha\,1}^2,\partial\mathbf{Z}(\mathbf{x}_{\beta\,1})/\partial y} & \cdots & \mathbf{C}_{\mathbf{x}_{\alpha\,i}^2,\partial\mathbf{Z}(\mathbf{x}_{\beta\,j})/\partial z} \\ \vdots & \vdots & \ddots & \vdots & \ddots & \vdots \\ \mathbf{C}_{\mathbf{x}_{\alpha\,1}^r,\partial\mathbf{Z}(\mathbf{x}_{\beta\,1})/\partial x} & \mathbf{C}_{\mathbf{x}_{\alpha\,2}^r,\partial\mathbf{Z}(\mathbf{x}_{\beta\,1})/\partial x} & \cdots & \mathbf{C}_{\mathbf{x}_{\alpha\,2}^r,\partial\mathbf{Z}(\mathbf{x}_{\beta\,1})/\partial y} & \cdots & \mathbf{C}_{\mathbf{x}_{\alpha\,i}^r,\partial\mathbf{Z}(\mathbf{x}_{\beta\,j})/\partial z} \end{bmatrix}$$

(B17)

As the interfaces are relative to a point $\mathbf{x}_{\alpha\,0}^k$ the value of the covariance function:

$$\mathbf{C}_{\mathbf{x}_{\alpha\,i}^r,\partial\mathbf{Z}(\mathbf{x}_{\beta\,j})/\partial x} = C_{Z(\mathbf{x}_{\alpha\,i}^r),\partial Z(\mathbf{x}_{\mathbf{fi}j})/\partial x} - C_{Z(\mathbf{x}_{\alpha\,0}^r),\partial Z(\mathbf{x}_{\mathbf{fi}j})/\partial x}$$

(B18)

with the covariance of the scalar field being function the vector r, its directional derivative is analogous to the previous derivations:

$$\mathbf{C}_{\mathbf{Z},\partial\mathbf{Z}/\partial\mathbf{u}} = \frac{\partial C_{\mathbf{Z}}(r)}{\partial r}\frac{\partial r}{\partial h_u} = -\frac{h_u}{r}C_Z'$$

(B19)

**B4    Universal matrix**

As the mean value of the scalar field is going to be always unknown, it needs to be estimated from data itself. The simplest approach is to consider the mean constant for the whole domain, i.e. ordinary Kriging. However, in the *scalar field* case we can assume certain drift towards the direction of the orientations. Therefore, the mean can be written as function of known basis functions:

$$\mu(\mathbf{x}) = \sum_{l=0}^{L} a_l f^l(\mathbf{x})$$

(B20)

where $l$ is the grade of the polynomials used to describe the drift. Because of the algebraic dependence of the variables, there is only one drift and therefore the unbiasedness can be expressed as:

$$\mathbf{U}_{\mathbf{Z}}\lambda_1 + \mathbf{U}_{\partial\mathbf{Z}/\partial u}\lambda_2 = f_{10}$$

(B21)

Consequently, the number of equations are determined according to the grade of the polynomial and the number of equations forming the properties matrices equations B15 and B1:

$$U_Z = \begin{bmatrix} x_1^1 - x_0^1 & x_2^1 - x_0^1 & \cdots & x_1^2 - x_0^2 & x_2^2 - x_0^2 & \cdots & x_{i-1}^r - x_0^r & x_i^r - x_0^r \\ y_1^1 - y_0^1 & y_2^1 - y_0^1 & \cdots & y_1^2 - y_0^2 & y_2^2 - y_0^2 & \cdots & y_{i-1}^r - y_0^r & y_i^r - y_0^r \\ z_1^1 - z_0^1 & z_2^1 - z_0^1 & \cdots & z_1^2 - z_0^2 & z_2^2 - z_0^2 & \cdots & z_{i-1}^r - z_0^r & z_i^r - z_0^r \\ x_1^1 x_1^1 - x_0^1 x_0^1 & x_2^1 x_2^1 - x_0^1 x_0^1 & \cdots & x_1^2 x_1^2 - x_0^2 x_0^2 & x_2^2 x_2^2 - x_0^2 x_0^2 & \cdots & x_{i-1}^r x_{i-1}^r - x_0^r x_0^r & x_i^r x_i^r - x_0^r x_0^r \\ y_1^1 y_1^1 - y_0^1 y_0^1 & y_2^1 y_2^1 - y_0^1 y_0^1 & \cdots & y_1^2 y_1^2 - y_0^2 y_0^2 & y_2^2 y_2^2 - y_0^2 y_0^2 & \cdots & y_{i-1}^r y_{i-1}^r - y_0^r y_0^r & y_i^r y_i^r - y_0^r y_0^r \\ z_1^1 z_1^1 - z_0^1 z_0^1 & z_2^1 z_2^1 - z_0^1 z_0^1 & \cdots & z_1^2 z_1^2 - z_0^2 z_0^2 & z_2^2 z_2^2 - z_0^2 z_0^2 & \cdots & z_{i-1}^r z_{i-1}^r - z_0^r z_0^r & z_i^r z_i^r - z_0^r z_0^r \\ x_1^1 y_1^1 - x_0^1 y_0^1 & x_2^1 y_2^1 - x_0^1 y_0^1 & \cdots & x_1^2 y_1^2 - x_0^2 y_0^2 & x_2^2 y_2^2 - x_0^2 y_0^2 & \cdots & x_{i-1}^r y_{i-1}^r - x_0^r y_0^r & x_i^r y_i^r - x_0^r y_0^r \\ x_1^1 z_1^1 - x_0^1 z_0^1 & x_2^1 z_2^1 - x_0^1 z_0^1 & \cdots & x_1^2 z_1^2 - x_0^2 z_0^2 & x_2^2 z_2^2 - x_0^2 z_0^2 & \cdots & x_{i-1}^r z_{i-1}^r - x_0^r z_0^r & x_i^r z_i^r - x_0^r z_0^r \\ y_1^1 z_1^1 - y_0^1 z_0^1 & y_2^1 z_2^1 - y_0^1 z_0^1 & \cdots & y_1^2 z_1^2 - y_0^2 z_0^2 & y_2^2 z_2^2 - y_0^2 z_0^2 & \cdots & y_{i-1}^r z_{i-1}^r - y_0^r z_0^r & y_i^r z_i^r - y_0^r z_0^r \end{bmatrix}$$

(B22)

$$\mathbf{U}_{\partial\mathbf{Z}/\partial\mathbf{u}} = \begin{array}{cc} \begin{array}{cccccccc} \mathbf{x_{fi1}} & \mathbf{x_{fi2}} & ... & \mathbf{x_{fi1}} & \mathbf{x_{fi2}} & ... & \mathbf{x_{fii-1}} & \mathbf{x_{fii}} \end{array} & \\ \begin{bmatrix} 1 & 1 & ... & 0 & 0 & ... & 0 & 0 \\ 0 & 0 & ... & 1 & 1 & ... & 0 & 0 \\ 0 & 0 & ... & 0 & 0 & ... & 1 & 1 \\ 2x_1 & 2x_2 & ... & 0 & 0 & ... & 0 & 0 \\ 0 & 0 & ... & 2y_1 & 2y_2 & ... & 0 & 0 \\ 0 & 0 & ... & 0 & 0 & ... & 2z_{i-1} & 2z_i \\ y_1 & y_2 & ... & x_1 & x_2 & ... & 0 & 0 \\ y_1 & y_2 & ... & 0 & 0 & ... & x_{i-1} & x_i \\ 0 & 0 & ... & z_1 & z_2 & ... & x_{i-1} & x_i \end{bmatrix} & \begin{array}{l} \partial\mathbf{x_{fi}}/\partial x \\ \partial\mathbf{x_{fi}}/\partial y \\ \partial\mathbf{x_{fi}}/\partial z \\ \partial^2\mathbf{x_{fi}}/\partial x^2 \\ \partial^2\mathbf{x_{fi}}/\partial y^2 \\ \partial^2\mathbf{x_{fi}}/\partial z^2 \\ \partial^2\mathbf{x_{fi}}/\partial x\partial y \\ \partial^2\mathbf{x_{fi}}/\partial x\partial z \\ \partial^2\mathbf{x_{fi}}/\partial y\partial z \end{array} \end{array} \tag{B23}$$

**Appendix C: Kriging Estimator**

In normal Kriging the right hand term of the Kriging system (Eq. 4) corresponds to covariances and drift matrices of dimensions $m \times n$ where $m$ is the number of elements of the data sets—either $\mathbf{x}_\alpha$ or $\mathbf{x}_\beta$—and $n$ the number of locations where the interpolation is performed, $\mathbf{x}_0$.

Since, in this case, the parameters of the variogram functions are arbitrarily chosen, the Kriging variance does not hold any physical information of the domain. As a result of this, being interested

only in the mean value, we can solve the Kriging system in the dual form (Matheron, 1981):

$$Z(\mathbf{x}_0) = \begin{bmatrix} a'_{\partial\mathbf{Z}/\partial u,\,\partial\mathbf{Z}/\partial v} & b'_{\mathbf{Z},\,\mathbf{Z}} & c' \end{bmatrix} \begin{bmatrix} \mathbf{c}_{\partial\mathbf{Z}/\partial\mathbf{u},\,\partial\mathbf{Z}/\partial\mathbf{v}} & \mathbf{c}_{\partial\mathbf{Z}/\partial\mathbf{u},\,\mathbf{Z}} \\ \mathbf{c}_{\mathbf{Z},\,\partial\mathbf{Z}/\partial\mathbf{u}} & \mathbf{c}_{\mathbf{Z},\,\mathbf{Z}} \\ \mathbf{f_{10}} & \mathbf{f_{20}} \end{bmatrix} \tag{C1}$$

where:

$$\begin{bmatrix} a_{\partial\mathbf{Z}/\partial u,\,\partial\mathbf{Z}/\partial v} \\ b_{\mathbf{Z},\,\mathbf{Z}} \\ c \end{bmatrix} = \begin{bmatrix} \partial\mathbf{Z} \\ 0 \\ 0 \end{bmatrix} \begin{bmatrix} \mathbf{C}_{\partial\mathbf{Z}/\partial\mathbf{u},\,\partial\mathbf{Z}/\partial\mathbf{v}} & \mathbf{C}_{\partial\mathbf{Z}/\partial\mathbf{u},\,\mathbf{z}} & \mathbf{U}_{\partial\mathbf{Z}/\partial\mathbf{u}} \\ \mathbf{C}_{\mathbf{Z},\,\partial\mathbf{Z}/\partial\mathbf{u}} & \mathbf{C}_{\mathbf{Z},\,\mathbf{Z}} & \mathbf{U}_{\mathbf{Z}} \\ \mathbf{U}'_{\partial\mathbf{Z}/\partial\mathbf{u}} & \mathbf{U}'_{\mathbf{Z}} & \mathbf{0} \end{bmatrix}^{-1} \tag{C2}$$

noticing that the 0 on the second row appears due to we are interpolation the difference of scalar

fields instead the scalar field itself 2.

**Appendix D: Example of covariance function: cubic**

The choice of the covariance function will govern the shape of the iso-surfaces of the scalar field. As opposed to other Kriging uses, here the choice cannot be based on empirical measurements. Therefore, the choice of the covariance function is merely arbitrary trying to mimic as far as possible

coherent geological structures.


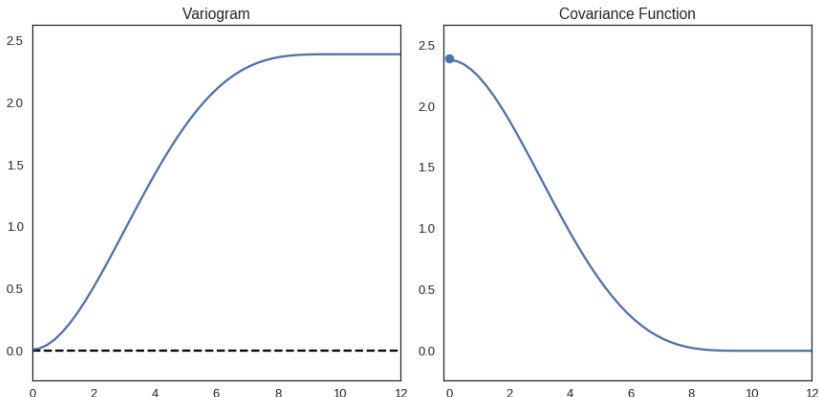

**Figure 12.** Representation of a cubic variogram and covariance for an arbitrary range and nugget effect.

The main requirement to take into consideration when the time comes to choose a covariance function is that it has to be twice differentiable, $h^2$ in origin to be able to calculate $\mathbf{C}_{\partial\mathbf{Z}/\partial\mathbf{u},\,\partial\mathbf{Z}/\partial\mathbf{v}}$ as we saw in equation B13. The use of a Gaussian model $C(r) = \exp -(r/a)^2$ and the non-divergent spline $C(r) = r^4 Log(r)$ and their correspondent flaws are explored in Lajaunie et al. (1997).

The most widely used function in the potential field method is the cubic covariance due to mathematical robustness and its coherent geological description of the space.

$$C(r) = \begin{cases} C_0 \left(1 - 7(\frac{r}{a})^2 + \frac{35}{4}(\frac{r}{a})^3 - \frac{7}{2}(\frac{r}{a})^5 + \frac{3}{4}(\frac{r}{a})^7\right) & \text{for } 0 \leq r \leq a \\ 0 & \text{for } r \geq a \end{cases} \tag{D1}$$

with $a$ being the range and $C_0$ the variance of the data. The value of $a$ determine the maximum distance that a data point influence another. Since, we assume that all data belong to the same depositional phase it is recommended to choose values close to the maximum extent to interpolate in order to avoid mathematical artifacts. for the values of the covariance at 0 and nuggets effects so far only *ad hoc* values have been used so far. It is important to notice that the only effect that the values of the covariance in the potential-field method has it is to weight the relative influence of both CoKriging parameters (interfaces and orientations) since te absolut value of the field is meaningless. Regarding the nugget effect, the authors recommendation is to use fairly small nugget effects to give stability to the computation—since we normally use the kriging mean it should not have further impact to the result.

**Appendix E:  Probabilistic Graphical Model**

Here we can see the probabilistic graphical model of the Bayesian inference of Section 3.4.2:





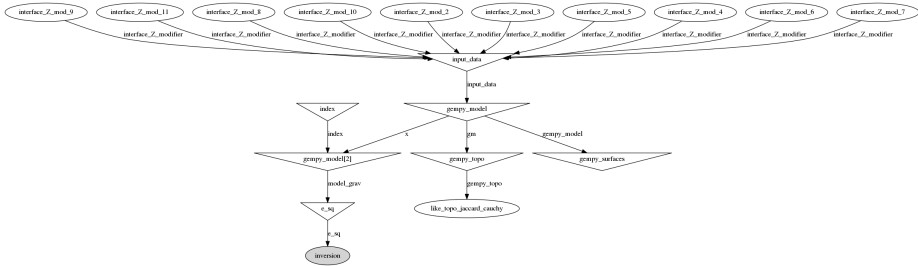

**Figure 13.** Probabilistic graphical model generated with pymc2. Ellipses represent stochastic parameters, while triangles are deterministic functions that return intermediated states of the probabilistic model such as the GemPy model

## Appendix F: Blender integration

Along the paper we have mentioned and show Blender visualizations (figure 1, b). The first step to obtain them is to be able to run *GemPy* in Blender's integrated Python (there are several tutorials online to use external libraries in Blender). Once it is running, we can use Blender's library *bpy* to generate Blender's actors directly from code. Here We include the code listing with the extra functions necessary to create automatically *GemPy* models in Blender

**Listing 8.** Extra functionality needed to create GemPy models in Blender

```
# Import Blender library, GemPy and GemPy colors
import bpy
import gempy as gp
from gempy.colors import color_lot

# Delete previous objects
try:
        bpy.ops.object.mode_set(mode='OBJECT')
        bpy.ops.object.select_by_type(type='MESH')
        bpy.ops.object.delete(use_global=False)
        for item in bpy.data.meshes:
                bpy.data.meshes.remove(item)
except:
        pass

# Define functio to create a Blender material, i.e. texture
def makeMaterial(name, diffuse, specular, alpha):
        mat = bpy.data.materials.new(name)
        mat.diffuse_color = diffuse
        #mat.diffuse_shader = LAMBERT
        mat.diffuse_intensity = 1.0
        mat.specular_color = specular
        #    mat.specular_shader = COOKTORR
        mat.specular_intensity = 0.5
        mat.alpha = alpha
        mat.ambient = 1
```



```
              return mat
      # Define function the assing material to object
      def setMaterial(ob, mat):
              me = ob.data
              me.materials.append(mat)
      ...
      add listing 1
      ...

# Create interface spheres and orientation copes out of a (rescaled) geodata
      # Interfaces
      for e, val in enumerate(interp_data.geo_data_res.interfaces.iterrows()):
              index = val[0]
              row = val[1]
color = makeMaterial('color',color_lot[row['formation_number']],(1,1,1),1)
              origin = (row['X']*10, row['Y']*10, row['Z']*10)
              bpy.ops.mesh.primitive_uv_sphere_add(location=origin, size=0.1)
              setMaterial(bpy.context.object, color)

# Orientations
      for e, val in enumerate(interp_data.geo_data_res.orientations.iterrows()):
              index = val[0]
              row = val[1]
              red = makeMaterial('Red',color_lot[row['formation_number']],(1,1,1),1)
origin = (row['X']*10, row['Y']*10, row['Z']*10)
              rotation_p = (row['G_y'], row['G_x'], row['G_z'])
              bpy.ops.mesh.primitive_cone_add(location=origin, rotation=rotation_p)
              bpy.context.object.dimensions = [.3,.3,.3]
              #bpy.ops.transform.translate(value=(1,0,0))
setMaterial(bpy.context.object, red)

      # Create rescaled simpleces
      verts, faces = gp.get_surfaces(interp_data,lith_block[1],
                                     fault_block[1],
original_scale=False)
      # or import them from a previous project
      import numpy as np
      verts = np.load('perth_ver.npy')
      faces = np.load('perth_sim.npy')
      # Create surfaces
      for i in range(0,n_formations):
              mesh_data = bpy.data.meshes.new("cube_mesh_data")
              mesh_data.from_pydata(verts[i]*10, [], faces[i].tolist())
mesh_data.update()

              obj = bpy.data.objects.new("My_Object", mesh_data)
              red = makeMaterial('Red',color_lot[i+1],(1,1,1),1)
              setMaterial(obj, red)
scene = bpy.context.scene
              scene.objects.link(obj)
              obj.select = True
```





*Acknowledgements.* The authors would like to acknowledge all people—all around the world—who has con-

tributed to the final state of the library either by stimulating mathematical discussions or finding *bugs*. This
project could not be possible without the invaluable help of all them.



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
