# Peer review of "GemPy 1.0: open-source stochastic geological modeling and inversion"

_Geoscientific Model Development, 2018_

## Referee Comment (RC1) · Anonymous Referee #1 · 8 May 2018

Gempy.bib

[a4paper]article [english]babel [utf8x]inputenc [T1]fontenc

[a4paper,top=3cm,bottom=2cm,left=3cm,right=3cm,marginparwidth=1.75cm]geometry

url amsmath graphicx [colorinlistoftodos]todonotes [colorlinks=true, allcolors=blue]hyperref amssymb siunitx tabularx algorithm,algorithmicx amsmath,algpseudocodealgcompatible amsthm graphicx lscape verbatim color,soul xcolor subfigure tabularx,ragged2e,booktabs,caption,array,multirow,multicol csquotes mathtools lineno amsthm listings amssymb latexsym epsfig float xspace float

**Review of Manuscript**
**GemPy 1.0: open source stochastic geological modeling and inversion**

May 8, 2018

The aim of this paper is to develop a framework for fusing many sources of information into a coherent probablistic model to allow for estimation and inference in geological inversion problems. It is an ambitious task and the authors are to be commended on tackling it, and making code publically available. I am not an expert in geological inversion problems, although I have some familiarity with them, so I will confine my comments to a discussion on the probabilistic model, Bayesian inference and probabilistic programming sections.

1. The paper would benefit from a clear and concise description of an example of at least one probabilistic model.

    (a) For example in section **3.4.2 Geological Inversion: Gravity and Topology** the authors say that they construct a specific likelihood function for a topology, but no likelihood function is given. The authors correctly state that the Jaccard index varies between 0 and 1, but then go on to state that it is *a single number we can evaluate using a a probability density function. The type of probability density function used will determine the strength or likelihood that the mean graph represent.* What does this sentence mean? They go on to say *Here we use a half Cauchy, due to tolerance for outliers.* Why? A half Cauchy has support on the interval $[0, \infty)$, whereas this statistic has support on the interval $[0, 1]$. What is meant by *its tolerance for outliers*?

(b) What is needed is a joint likelihood function on both the topology and the gravity to be specifically stated. See for example () and (). The authors should at least reference ().

(c) The authors statement *The use of likelihood functions in a Bayesian inference in opposition to simply rejection sampling has been explored by the authors during the recent years (de la Varga and Wellmann, 2016; Wellmann et al., 2017; Schaaf, 2017).* is confusing. Are the authors referring to likelihood free methods such as Approximate Bayesian Computation, ABC, where rejection sampling can be used to obtain draws from the approximate posterior? The use of likelihood function in Bayesian inference is typically not related to rejection sampling. Rejection sampling is a method to obtain draws from a non-standard distribution, in this case the posterior distribution, usually for the purpose of numerical integration. A likelihood is an assumption about the data generation process which, together with the prior, result in inference via the posterior. If the likelihood is unavailable in closed form, or if we do not wish to make assumptions about the data generating process, then the issue of how to approximate the posterior may involve rejection sampling. The authors need to articulate clearly the point they are making and provide a justification.

(d) A gravity likelihood is referred to on page 31. What is this likelihood? Are the authors assuming that the *observed* data is related to the *simulated*

data as a signal plus noise model of the form, $y_i = g(x_i) + e_i$, where $e_i$ is independently and identically distributed (i.i.d)? If so why do they model $(y_i - g(x_i))^2$ as a folded Cauchy (i.e a folded $t_1$)? What is this saying about the data generating process? Surely there is geophysical knowledge about the distribution of gravity measurements? From a statistical point of view gravity is an integral, a sum of things, in which case the central limit theorem (CLT) would make the assumption of Gaussian errors, i.e. $e_i \sim N(0, \sigma^2)$, reasonable. If this were so then and the observations independent (which I'm not convinced they would be), then $\sum_{i=1}^{n} (y_i - g(x_i))^2 \sim \chi_n^2$. Perhaps this is what they do, but it is not clear from the paper.

2. **MCMC convergence** The authors need to show that the MCMC scheme converges. Convergence in geophysical inversion problems is non trivial. Posterior distributions of geophysical inversion problems are notoriously difficult to explore, for a discussion see (). and for a demonstration of how difficult they are to explore see (). The NUTS algorithm used in python works well when the derivative exists and is well behaved, but as the posterior distribution in () shows, these distributions can have many modes and derivatives which are difficult, if not impossible to compute. Parallel tempering is probably the best way to explore these multi-model distributions, as shown in ().

3. Minor points

   - The Jaccard index given by equation 13 is not a likelihood function, nor, as it is written, is it even a measure. The authors correctly state that the Jaccard index is a statistic used to compare sets, in this case topologies. It is the ratio of the *size* of the intersection over *size* of the union. It should be written as

   $$J(A, B) = \frac{|A \cap B|}{|A \cup B|}$$

   where the notation $|.|$ denotes a measure of size to be defined.

- change the phrase *due to tolerance for outliers* to *because parameter estimates based on Cauchy likelihoods are more robust to outliers than parameter estimates based on, say, Gaussian likelihoods*.

**References**

Beardsmore, G and Durrant-Whyte, H and McCalman, L and O'Callaghan, S and Reid, A. (2016) *A Bayesian Inference Tool for Geophysical Joint Inversions*, ASEG Extended Abstracts 2016: 25th International Geophysical Conference and Exhibition, 509–518. https://library.seg.org/doi/abs/10.1071/ASEG2016ab089.

Beskos, A. and Girolami, M. and Lan, S. and Farrell, E. and Stuart, M.(2017) *Geometric MCMC for infinite-dimensional inverse problems*, Journal of Computational Physics, 335, pp 327-351

Chandra, R and Azam, D and Muller, D and Lasalles T and Cripps, S. (2018) *Bayesian Dynamic Earth models, landscape dynamics and basin evolution*, In Review (Computers and Geoscience), https://github.com/rohitash-chandra/research/blob/master/2018/BayesLands.pdf

---

## Referee Comment (RC2) · Anonymous Referee #2 · 8 Jun 2018

Review of Manuscript gmd-2018-61 entitled "GemPy 1.0: open-source stochastic geological modeling and inversion" by Miguel da la Varga, Alexander Schaaf, and Florian Wellmann

Dear Editor and Authors,

This manuscript presents an open source implicit geomodelling method implemented in python that is capable of generating 3D geological models with complex lithostratigraphic structures, fault networks, and unconformities. Currently, there are no open source solutions that exist with all of these comprehensive features. In addition, gempy provides all the necessary tools to perform complex 3D modelling, visualization, and analysis out of the box and provides a much needed ecosystem for scientific research permitting enhancement of existing methodologies as well as potential addition of new

methods, techniques, and tools benefitting a suite of various geoscience applications. The fact that their method can be integrated into machine learning and Bayesian inference frameworks for stochastic modelling and inversion is indeed exciting and opens up numerous possibilities. I have many suggestions to improve the paper's descriptions and clarity. I recommend this manuscript for publication after the authors have addressed the issues noted in the review.

Specific Comments

My detailed comments and edits can be found with the attached pdf. Some general comments are given below.

The sections regarding Bayesian inference/Probabilistic modelling is weak, in my opinion. Although, it should be noted that the reviewer's expertise does not lie within this domain but within mathematical approximation and the implicit approach. The general formulation of Bayesian inference and how it is integrated with the variables of the interpolants/estimators from the implicit approach should be presented. The Appendix E needs to be expanded and described in more detail. In addition, there is a lot of Bayesian/probability nomenclature not properly defined in section 2.3. The manuscript would benefit from clearly stated definitions.

It's unclear, to me at least, how fault drift functions are chosen or designed and how the form of that function affects the modelled results. Is it a trial and error process – finding the result that maximizes the sharpness on the transitions around the discontinuities? Perhaps an appendix section would be useful for this. In addition, I find all python code samples involving faults confusing (e.g. Listing 3). For example, there is no input data describing the fault (location and orientation points) how can a fault's scalar field be produced without this information? Only the order of the formations involving the fault is given.

The work entailed in the manuscript has a high potential for establishing a research community for developing next-generation geological modelling algorithms. There can

be many improvements made to the current code base but what has been completed is impressive.

Please also note the supplement to this comment:
https://www.geosci-model-dev-discuss.net/gmd-2018-61/gmd-2018-61-RC2-supplement.pdf

—————————————————————

[Figure]

**Supplement:**

Manuscript prepared for Geosci. Model Dev. with version 2015/04/24 7.83 Copernicus papers of the LATEX class copernicus.cls. Date: 28 February 2018

**GemPy 1.0: open-source stochastic geological modeling and inversion**

Miguel de la Varga, Alexander Schaaf, and Florian Wellmann Institute for Computational Geoscience and Reservoir Engineering Correspondence to: varga@aices.rwth-aachen.de

Abstract. The representation of subsurface structures is an essential aspect of a wide variety of geoscientific investigations and applications: ranging from geofluid reservoir studies, over raw material investigations, to geosequestration, as well as many branches of geoscientific research studies and applications in geological surveys. A wide range of methods exists to generate geological models.

- However, especially the powerful methods are behind a paywall in expensive commercial packages. 5 We present here a full open-source geomodeling method, based on an implicit potential-field interpolation approach. The interpolation algorithm is comparable to implementations in commercial packages and capable of constructing complex full 3-D geological models, including fault networks, fault-surface interactions, unconformities, and dome structures. This algorithm is implemented in
- the programming language Python, making use of a highly efficient underlying library for efficient 10 code generation (theano) that enables a direct execution on GPU's. The functionality can be separated into the core aspects required to generate 3-D geological models and additional assets for advanced scientific investigations. These assets provide the full power behind our approach, as they enable the link to Machine Learning and Bayesian inference frameworks and thus a path to stochas-
- tic geological modeling and inversions. In addition, we provide methods to analyse model topology 15 and to compute gravity fields on the basis of the geological models and assigned density values. In summary, we provide a basis for open scientific research using geological models, with the aim to foster reproducible research in the field of geomodeling.

**Contents**

[revised manuscript text omitted]

```

---

## Author Comment (AC1) · 6 Jul 2018

**Answer Review 1:**

Thank you very much for your suggestions, detailed comments and questions, as well as the positive feedback. We carefully revised the manuscript and provide below a detailed reply to all comments. We also attached a pdf with highlighted changes between the original submission and the revised version.

We agree with the reviewer that the explanation of the Bayesian network was not clear and probably insufficient. However, the scope of this particular paper was on the gen-

erative model of the Bayesian inference. Since the whole library has been written to be part of a Bayesian inference, we consider it appropriate to just to show a vertical slice of what is possible. Nevertheless, we have reworked quite that part of the paper quite a bit and now include the whole probabilistic graphical model into the main text in order to be more consistent and clear. Also, we have added the convergence analysis in the appendix.

Regarding the reviewers general concerns about the construction of the likelihood functions and samplers, we shared many of them and we are working hard to find the optimal Bayesian model for different data sets. The comments were very insightful and hopefully we will be able to explore them in a more focused paper in the future.

**RESPONSES**

**[1]** The paper would benefit from a clear and concise description of an example of at least one probabilistic model.

**Authors response:** We have extended and moved to the main text the specific probabilistic graphical model

**Change in manuscript:**

**[1a]** For example in section 3.4.2 Geological Inversion: Gravity and Topology the authors say that they construct a specific likelihood function for a topolC2 GMDD Interactive comment Printer-friendly version Discussion paper ogy, but no likelihood function is given. The authors correctly state that the Jaccard index varies between 0 and 1, but

then go on to state that it is a single number we can evaluate using a a probability density function. The type of probability density function used will determine the strength or likelihood that the mean graph represent. What does this sentence mean? They go on to say Here we use a half Cauchy, due to tolerance for outliers. Why? A half Cauchy has support on the interval [0, $\infty$ ), whereas this statistic has support on the interval [0, 1]. What is meant by its tolerance for outliers?

**Authors response:** We use a half-Cauchy distribution to evaluate the likelihood of the Jaccard index of the simulated model topology due to its wider tail. It is not used directly as a likelihood, but rather as a factor potential which allows us to incorporate the "soft data" of our topology information into the Bayesian inference. As the Jaccard index results in values within the interval $[0,1]$ , the half-Cauchy function is only evaluated for those given values. The shape parameter $\beta$ was chosen empirically, as it showed promising results for effective parameter space exploration in the used MCMC scheme.

**Change in manuscript:** *To evaluate the likelihood of the simulated model topology we use a factor potential with a half-Cauchy parametrization (shape parameter $\alpha=0$ and rate parameter $\beta==10^{-3}$ ) to constrain our model using the "soft data" of our topological knowledge \citep{ lauritzen1990independence, jordan1998learning, christakos2002assimilation} . This specific parametrization was chosen due to empirical evidence from different model runs to allow for effective parameter space exploration in the used MCMC scheme.*

**[1b]** What is needed is a joint likelihood function on both the topology and the gravity to be specifically stated. See for example () and (). The authors should at least reference ().

**Authors response:** In pymc when you specify more than one likelihood/potential, it automatically starts sampling on the joint likelihood.

**Change in manuscript: [1010]** *Defining the topology potential and gravity likelihood on the same Bayesian network creates a join likelihood value that we need to sample from*

**[1c]** The authors statement The use of likelihood functions in a Bayesian inference in opposition to simply rejection sampling has been explored by the authors during the recent years (de la Varga and Wellmann, 2016; Wellmann et al., 2017; Schaaf, 2017). is confusing. Are the authors referring to likelihood free methods such as Approximate Bayesian Computation, ABC, where rejection sampling can be used to obtain draws from the approximate posterior? The use of likelihood function in Bayesian inference is typically not related to rejection sampling. Rejection sampling is a method to obtain draws from a non-standard distribution, in this case the posterior distribution, usually for the purpose of numerical integration. A likelihood is an assumption about the data generation process which, together with the prior, result in inference via the posterior. If the likelihood is unavailable in closed form, or if we do not wish to make assumptions about the data generating process, then the issue of how to approximate the posterior may involve rejection sampling. The authors need to articulate clearly the point they are making and provide a justification.

**Authors response:** In the sentence we were confusing rejection sampling as a parameter space exploration method to approximate a posterior with just forward simulating the priors. We agree with the reviewers comments and adjusted the sentence in line 855.

**Change in manuscript [855]**: "*The use of likelihood functions in a Bayesian inference in comparison to simple forward simulation has been explored by the authors during recent years (de la Varga and Wellmann, 2016; Wellmann et al., 2017; Schaaf, 2017).*"

**[1d]** A gravity likelihood is referred to on page 31. What is this likelihood? Are the authors assuming that the observed data is related to the simulated data as a signal plus noise model of the form, $y_i = g(x_i) + e_i$ , where $e_i$ is independently and identically distributed (i.i.d)? If so why do they model $(y_i - g(x_i))2$ as a folded Cauchy (i.e a folded $t_1$)? What is this saying about the data generating process? Surely there is geophysical knowledge about the distribution of gravity measurements? From a statistical point of view gravity is an integral, a sum of things, in which case the central limit theorem (CLT) would make the assumption of Gaussian errors, i.e. $e_i \sim N(0, \sigma 2)$, reasonable. If this were so then and the observations independent (which I'm not convinced they would be), then $P^n_{i=1}$ $(y_i - g(x_i))2 \sim \chi 2 n$ . Perhaps this is what they do, but it is not clear from the paper

**Authors response**: We agree with the reviewer that the explanation was not sufficiently clear, in part because it was not the main purpose of this paper and in part because we did not find yet a convincing way to construct the likelihood function. The model suggested by the reviewer is quite close to what we presented here and we agree with his concerns about the correlation but again we prefer to focus around the generative model since the paper is already too large. Nevertheless, e have extended the explanation of this part and adding the complete PGM figure into the main body text

**[2]** MCMC convergence The authors need to show that the MCMC scheme converges. Convergence in geophysical inversion problems is non trivial. Posterior

distributions of geophysical inversion problems are notoriously difficult to explore, for a discussion see (). and for a demonstration of how difficult they are to explore see (). The NUTS algorithm used in python works well when the derivative exists and is well behaved, but as the posterior distribution in () shows, these distributions can have many modes and derivatives which are difficult, if not impossible to compute. Parallel tempering is probably the best way to explore these multi-model distributions, as shown in ().

**Authors response**: We added a traces plot and a Geweke test into the appendix. Again, we share the concerns about the samplers and we are actively studying the best combinations of them for this type of problems. We agree that a combination of gradient based and parallel methods could be the best solution in the middle term. However, we also consider that that is beyond the scope of this publication.

**[3a]** The Jaccard index given by equation 13 is not a likelihood function, nor, as it is written, is it even a measure. The authors correctly state that the Jaccard index is a statistic used to compare sets, in this case topologies. It is the ratio of the size of the intersection over size of the union. It should be written as

$$J(A, B) = |A \cap B| / |A \cup B|$$

where the notation $|.|$ denotes a measure of size to be defined.

**Authors response:** Adjusted Jaccard index to correctly contain the absolutes. Thanks for catching this!

**Change in manuscript:** Changed Eq. 13 to J(A, B) = | A ∩ B| / | A ∪ B|

**[3b]** change the phrase due to tolerance for outliers to because parameter estimates based on Cauchy likelihoods are more robust to outliers than parameter estimates based on, say, Gaussian likelihoods.

**Authors response:** We agree to the change in wording, but adjusted the whole paragraph to more precisely state the use of the topology information as a factor potential in the Bayesian inference and the second reviewers comments on line [811].

**Change in manuscript:** *To evaluate the likelihood of the simulated model topology we use a factor potential with a half-Cauchy parametrization (shape parameter $ \ alpha=0$ and rate parameter $ \ beta==10ˆ { -3} $ ) to constrain our model using the "soft data" of our topological knowledge \ citep{ lauritzen1990independence, jordan1998learning, christakos2002assimilation} . This specific parametrization was chosen due to empirical evidence from different model runs to allow for effective parameter space exploration in the used MCMC scheme and due to the Cauchy distribution being more robust to outliers than parameter estimates based on, say, Gaussian likelihoods.*

---

## Author Comment (AC2) · 6 Jul 2018

**Answer Review 2:**

Thank you very much for the detailed suggestions and comments and questions, as well as the positive feedback. After incorporating your input, the paper has become much more polished and easier to understand.

Despite agreeing with most of the recommendations about expanding the explanations about the probabilistic graphical models and fault networks, we are the opinion that

the extent of the paper is already too large to incorporate this information. Therefore sections that may be self contained in further publications—i.e. Bayesian network, fault network—or that have been already published (although sadly quite sparsely) have been reworked but not extended. Hopefully all of this will come together in the first's author PhD thesis.

Regarding the Bayesian model, the scope of this particular paper was on the generative model of the Bayesian inference—i.e. the mathematical formulation that connects the model parameters with different datasets. We have improved the clarity of our description and added the probabilistic graphical model into the main text.

The definition of faults drift functions are, again, too broad to be treated in an already too large paper. So far the drift function is manually defined and in the authors view the best results would be yielded by optimization. We are not aware of work done in this direction and we have just started to explore this option. Regarding their definition in the GemPy code, we have added some further explanation in the manuscript but, in short, it is done by categorizing the input data. Although in the paper we aim to give a flavour of the use of GemPy, for a good understanding of how to create models, we recommend the online documentation and tutorials (an endless work-in-progress). We will try to provide an example of fault networks sooner rather than later.

Finally, we took the decision of writing this paper as a compendium of the necessary algebra with the correspondent open-source code to generate implicit geological models. We are aware that a more extensive explanation of the kriging system and the Appendix C would be useful for the community but our mathematical knowledge is limited and we are not convinced of being able to add much value to the already published work in this matter. Therefore we have opted for a more descriptive approach referencing the original sources.

Once again, we highly appreciate all comments and we accepted most suggestion not only for this paper but also for future work.

**RESPONSES**

Answer to pdf comments:

**[73]** reword

**Authors response:** reworded.

**Change in manuscript:** *The method was first introduce by \ cite{ Lajaunie.1997} and it is grounded on the mathematical principles of Universal CoKriging.*

**[102]** I would also say to provide an environment/ecosystem to improve/enhance/add existing methodologies.

**Authors response:** Accepted suggestion

**Change in manuscript:** *. . . , but to provide an environment to enhance existing methodologies as well as give access to an advanced modeling algorithm for scientific experiments in the field of geomodeling.*

**[135]** I wouldn't refer to this as anisotropy. More precisely, it describes the planar

orientation of the stratigraphic structure throughout the volume.

**Authors response:** Accepted suggestion. We agree, it is less confusing now

**Change in manuscript:** *The gradient of the scalar field will follow the planar orientation of the stratigraphic structure throughout the volume or, in other words...*

**[143]** reword

**Authors response:** Reword:

**Change in manuscript:** *...as it is not possible to obtain remotely accurate estimations*

**[168]** why 0 subscript? shouldn't the function 's variable be a general point x?

**Authors response:** Further explanation and added citation. x_0 is the terminology used by Chiles on his book Geostatistcs Modeling Spatial Uncertainty and the sub index 0 is used to differentiate the interpolated points x_0 to the input data x_alpha.

**Change in manuscript:** *where $ { \ bf{ x} } \_{ 0} $ refers to the estimated quantity for some integrable measure $ p\_0$ . . . . . we will try to be especially verbose regarding the mathematical terminology based primarily on \ cite{ Chiles.2004}*

**[173]** ?; unclear (and can lead to confusion) and not needed.

**Authors response:** Reword but not deleted. Despite we agree with the reviewer that the parameter p may lead to confusion we have decided to keep the original sentence for two reason:

1. Keep the nomenclature consistent with Chiles description

2. To point out that the method is multivariate and there is no limitations to extend the current mathematical formulation

**[184]** keep the terminology consistent, change to scalar field

**Authors response:** Potential field naming fixed, p substituted by rho and u defined as any unit vector. Everything here was carelessness from our side. Thank you to the reviewer for pointing them out.

**[188]** unclear

**Authors response:** Reword.

**Change in manuscript:** *Note that in this context the scalar field property $\ alpha$ is dimensionless. The only limitation is that the value must increase in the direction of the gradient which in turn describes the stratigraphic deposition.*

**[195]** Mention here that the choice of the reference point does not matter - choosing different reference points has no effect on the results.

**Authors response:** Added suggestion about reference point.

**Change in manuscript:** *It is important to mention that the choice of the reference points $\{ \bf{ x} \} _{ \alpha\ , 0} ^ k$ has no effect on the results.*

**[200]** most of this is unclear

**Authors response:** Improved explanation

**Change in manuscript:** *The advantage of this mathematical construction is that by not fixing the values of each interface $\{ \{ \bf{ Z} \} ( \bf{ x} \} _{ \alpha} ^ k )$ , the compression of layers—i.e. the rate of change of the scalar field—will be only defined by the gradients $\(\partial \{ \bf{ Z} \} / \partial u\ )$. This allows to propagate the effect of each gradient beyond the surrounding interfaces creating smoother formations.*

**[209]** define after this equation.

**Authors response:** Extended eq 3 definition

**[210]** Although this system is indeed correct. I am concerned that for most readers this system will be confusing. I would strongly suggest that appendix section C2 be expanded to explain in detail the meaning of this system and its clear consequences. Then from that derive separate linear systems for both the scalar field and the gradient of the scalar field. In addition, refer in lines 210-215 to that appendix section for full details.

**Authors response:** After long consideration. We have decided to leave it as system for two reason: (i) a proper explanation of a CoKriging system would need to be too long and we may not have the mathematical background to explain error-free, and (ii) the matricial form is more consistent—at least to us—with Chiles' book explanation.

[217] on the right you have a 3x2 matrix

**Authors response:** Fixed terminology

[220] this description is insufficient. e.g. $f_{10} = f_1(x_0)$ and you never defined what $x_0$ is. same goes for $f_{20}$. In connection with my comment about eqn 4 you should show or say why $f_{10}$ reduces to zero in the appendix. It would tremendously help new researchers fully understand the mathematics for potential enhancements - just as you have done for most of the kriging system.

**Authors response:** Added suggestions

**[300]** How do you ensure that the modelled conformable stratigraphic layers respect the specified sequence for the stratigraphic pile? e.g. the interpolation constraints for interface pts (the increment points) is that the scalar field at these interface pts is the same - the is no method of inputting the sequence via the increments since their particular value has no effect. This problem usually only presents itself when interface data is characterize by strong horizontally sampling bias.

**Authors response:** Added explanation. If we are not mistaken the reviewer refers here to the order of the formations within each series. Indeed this only depends on the

geometry and now we added a sentence clarifying this point

**Change in manuscript:** *It is interesting to point out that the the sequential pile only controls the order of each individual series. Within each series, the stratigraphic sequence is strictly determined by the geometry and the interpolation algorithm.*

**[345]** why is there no input data for the fault and the unconformity?

Added explanation. The separation between the different series/faults is done by labeling the input data. As the paper only contains a small snippet of the code this was indeed misleading. We increased the comments on the listing hoping to clarify this point.

**[370]** spelling

**Authors response:** Corrected

**[376]** which rules?

Reworded. We are aware that the fault section is a bit too vague but again the paper is already too long and the faulting algorithms may fit in future work.

**[379]** how does one choose the perfect drift function? trial and error?

**Authors response:** Essentially yes, i.e. optimization. Nobody so far has came out

with a better idea. Probably our framework would be the ideal place to test some of this since with AD optimizations work better.

**[391-392]** ?; unclear explicitly state what you mean by input parameters

**Authors response:** Reworded

**Change in manuscript:** *variables*

**[401]** explicitly define graph in the current context

**Authors response:** Reworded and added explanation of symbolic graph

**Change in manuscript:** *Writing symbolically requires a priori declaration of all algebra, from variables which will behave as latent parameters—i.e. the parameters we try to tune for optimization or*

*uncertainty quantification—to all involved constants and the specific mathematical functions that relates them. This statements generate a so called graph that encapsulates symbolically all the logic what enables to perform further analysis on the logic itself (e.g. differentiation or optimization).*

**[455]** colloquial. please reword. can a method have intuitions?

**Authors response:** Reworded

**[GMDD](GMDD)**

Interactive
comment

**Change in manuscript:** *There is extensive literature explaining in detail the method and its related intuitions since it*

**[498]** these have not yet been defined yet - as it relates to implicit modelling

**Authors response:** Added reference to chapter

**[FIGURE 6]** Very confusing figure

**Authors response:** Reworked in 3D to improve clarity

**[575]** what is this variable?

**Authors response:** Added exaplanation

**Change in manuscript:** $ t\_z$ —*i.e. the distance dependent side of Equation \ ref{ eq:grav\_0} —*

**[643]** how is this done?

**Authors response:** Added explanation to paragraph

**Change in manuscript:** *We multiply the binary fault array (0 for foot wall, 1 for hanging wall) with the maximum lithology value incremented by one. We then add it to the lithology array to make sure that layers that are in contact across faults are assigned a*

*unique integer in the resulting array.*

**[720]** Is there a best practices guide for PGM construction?

**Authors response:** Added some references. Sadly making Bayesian models nowadays is pretty much an art. The variability on topics datasets and complexity makes very difficult to give a close set of rules to construct the models. Probably the best reference would be Bayesian methods for hackers but seems to generalistic for the paper scope. Our past work and but in especial our future work will try to address this problem in a comprehensive manner since we agree that there is a lack guidelines especially in geological modeling

*Relevant citations with bibkeys:*

- { *bishop2013model*} *Bishop, C. M. (2013). Model-based machine learning. Phil. Trans. R. Soc. A, 371(1984), 20120222.*

- { *sucar2015probabilistic*} *Sucar, L. E. (2015). Probabilistic Graphical Models. Advances in Computer Vision and Pattern Recognition. London: Springer London. doi, 10, 978-1.*

- { *Patil:FseZoIYV*} *Patil, A., Huard, D., & Fonnesbeck, C. J. (2010). PyMC: Bayesian stochastic modelling in Python. Journal of statistical software, 35(4), 1.*

- { *Koller:2009wk*} *Koller, D., & Friedman, N. (2009). Probabilistic graphical models: principles and techniques. MIT press.*

**[734]** Why just the Z axis? It should be perturbed along each of the 3 axis. Does just choosing the one axis easier for generating results?

**Authors response:** Added explanation

*The choice of perturbing only the Z axis is merely due to computational limitations. Uncertainty tends to be higher in this direction (e.g. wells data or seismic velocity), however there is a lot of room for further research on the definition of prior data—i.e. its choice and probabilistic description—on both directions, to ensure that we properly explore the space of feasible models and to generate a parametric space as close as possible to the posterior.*

**[744]** reword

**Authors response:** Reworded

**[765]** what is the meaning of this variable?

**Authors response:** Removed . This is a pymc 2 dummy value to initialize the object. I just deleted it to avoid confusion.

**[FIGURE 8 CAPTION]** is this the number of realizations?; says probability of layer 2 in the figure! not layer 1

**Authors response:** Increased verbosity, fixed typo

**[850]** should change to p_i represents the probability of a layer at the i-th cell.

**Authors response:** Fixed typo

**[851]** compress? you mean quantity?

**Authors response:** Reworded

**Change in manuscript:** *we can use information entropy to reduce the dimensionality of probability fields into a single value at each voxel as an indication of uncertainty,*

**[856]** ? comparison

**Authors response:** Reworded

**[881]** what are these parameters? you also used alpha and beta elsewhere for something else

**Authors response:** Change in manuscript to specify parameters and how the topology is evaluated in the PGM:

**Change in manuscript:** *To evaluate the likelihood of the simulated model topology we use a factor potential with a half-Cauchy parametrization (shape parameter $\alpha=0$ and rate parameter $\beta==10^{-3}$ ) to constrain our model using the "soft data" of our topological knowledge \citep{ lauritzen1990independence, jor-*

*dan1998learning, christakos2002assimilation} . This specific parametrization was cho-
sen due to empirical evidence from different model runs to allow for effective parameter
space exploration in the used MCMC scheme and due to the Cauchy distribution being
more robust to outliers than parameter estimates based on, say, Gaussian likelihoods.*

**[948]** reword

**Authors response:** Reworded

*To sample from the posterior we use adaptive Metropolis*

**[958-961]** reword; Also which example ? from Fig 8's posterior models?

**Authors response:** Specified figure and reworded explanation

**[1018]** spelling

**Authors response:** Fixed

**[1036]** reword

**Authors response:** Fixed

**Change in manuscript:** *up to now*

**[1037]** reword

**Authors response:** Fixed

**Change in manuscript:** *able to construct*

**[1067]** general?

**Authors response:** Fixed

**Change in manuscript:** *general*

**[FIGURE 9]** Why are you presenting this graph? It does not add any insight. Please remove

**Authors response:** Reworked figures and equations. The reason to add this figures was to clarify the different distances that goes to each covariance functions since for us were one of the main challenges. However we agree that the implementation was not very clear. Hopefully after the rework the use of the figures is more obvious

**[1159]** please define hx, hy. shouldn't this be r = sqrt (hxˆ 2 + hyˆ 2 + hzˆ 2) with hx = xi - xj, hy = yi - yj, hz = zi - zj

**Authors response:** Added suggestion and reworded.

**Change in manuscript:**

**[GMDD](javascript:void)**
*\ begin{ equation}*

*r = \ sqrt{ hˆ 2_x+hˆ 2_y+hˆ 2_z}*

*\ end{ equation}*

*and $ h_u$ as the distance $ u_i - u_j$ in the given direction (usually Cartesian directions). Therefore, since we aim to derive $ C_{ Z} (r)$ respect an arbitrary direction $ u$ we must apply the \ textit{ directional derivative} rules as follows:*

**[1164]** repeat wrt lhs

**Authors response:** Fixed

**[1175]** reword; its related to the smoothness of a function

**Authors response:** Reworded and further explanation

**Change in manuscript:** *This derivation is independent to the covariance function of choice. However, some covariances may lead to mathematical indeterminations if they are not sufficiently differentiable.*

**[FIGURE 10]** Why are you presenting this graph? It does not add any insight. what co the different colored dots, lines, and arrows represent?

**Authors response:** same as Figure 9

**[1186]** keep terminology consistent throughout manuscript. change to scalar

Fixed

**[FIGURE 11]** Again, its unclear what the purpose of these graphs are trying to convey.

**Authors response:** same as figure 9

**[1194]** this eqn is incorrect. you have to be very careful with the notation. to properly show this you will have to restructure this.

**Authors response:** Fixed. Thank you for realizing. This was an important mistake.

**[1195]** ???

**Authors response:** Reworded.

**Change in manuscript:** *As the interfaces are relative to a reference point per later* $ { \ bf\{ x\} \} \_\{ \ alpha\_\ , 0\} ˆ k$
*the value of the covariance function will be the difference between this point and the rest on the same layer:*

**[1196]** formatting problem

**Authors response:** Fixed formatting

**[1214]** formatting errors. ; fii should be beta I assume

**Authors response:** Fixed formatting

**[1220]** Is there another reference for this? Matheron, 1981 is cryptic and written in a time (type writers) where there are no matrices in their explicit form. This reference is what is given in Lajaunie, but if there is a better reference can you add that?

**Authors response:** Added Chiles and Delfiner 2009 book reference

**[1235]** ??; these functions also have limits - especially so in scenarios where data sample highly variant geological structures.

**[1238]** before you indicated that C_0 is the value for the nugget effect.

**Authors response:** Fixed

**[1245-46]** grammar; not clear how one inputs the nugget effect given D1

**Authors response:** Added short explanation about nugget effect.

**Change in manuscript:** *The implementation of nugget effects in covariance matrices are done by adding the value to the diagonal.*

**[FIGURE 13]** Explain in more depth this figure. what are the "modifiers", why are there

11 of them?

**Authors response:** Whole PGM has been reworked manually and added to the main text.

Please also note the supplement to this comment:
https://www.geosci-model-dev-discuss.net/gmd-2018-61/gmd-2018-61-AC2-supplement.pdf

**Supplement:**

[revised manuscript text omitted]

---

## Author Response (AR2)

**Authors response:**

All the minor typos and wording have been addressed. Thank to the reviewer for finding them.

[revised manuscript text omitted]
} & \\ & 1 & 1 & \dots & 0 & 0 & \dots & 0 & 0 & \partial \mathbf{x}_{\beta i}/\partial x \\ & 0 & 0 & \dots & 1 & 1 & \dots & 0 & 0 & \partial \mathbf{x}_{\beta i}/\partial y \\ & 0 & 0 & \dots & 0 & 0 & \dots & 1 & 1 & \partial \mathbf{x}_{\beta i}/\partial z \\ & 2x_1 & 2x_2 & \dots & 0 & 0 & \dots & 0 & 0 & \partial^2 \mathbf{x}_{\beta i}/\partial x^2 \\ & 0 & 0 & \dots & 2y_1 & 2y_2 & \dots & 0 & 0 & \partial^2 \mathbf{x}_{\beta i}/\partial y^2 \\ & 0 & 0 & \dots & 0 & 0 & \dots & 2z_{i-1} & 2z_i & \partial^2 \mathbf{x}_{\beta i}/\partial z^2 \\ & y_1 & y_2 & \dots & x_1 & x_2 & \dots & 0 & 0 & \
[revised manuscript text omitted]